# Information-Theoretic Analysis of Unsupervised Domain Adaptation

## Abstract

This paper uses information-theoretic tools to analyze the generalization error in unsupervised domain adaptation (UDA). This study presents novel upper bounds for two notions of generalization errors. The first notion measures the gap between the population risk in the target domain and that in the source domain, and the second measures the gap between the population risk in the target domain and the empirical risk in the source domain. While our bounds for the first kind of error are in line with the traditional analysis and give similar insights, our bounds on the second kind of error are algorithm-dependent and also inspire insights into algorithm designs. Specifically, we present two simple techniques for improving generalization in UDA and validate them experimentally.

## 1 Introduction

This paper focuses on the *unsupervised domain adaptation (UDA)* task, where the learner is confronted with a source domain and a target domain and the algorithm is allowed to access to a labeled training sample from the source domain and an unlabeled training sample from the target domain. The goal is to find a predictor that performs well on the target domain.

A main obstacle in such a task is the discrepancy between the two domains. Some recent works have [1–9] proposed various measures to quantify such discrepancy, either for the UDA setting or for the more general domain generalization tasks, and many learning algorithms are proposed. For example, most recently, Nguyen et al. [9] uses a (reverse) KL divergence to measure the misalignment of the distributions of the two domains, and motivated by their generalization bound, they design an algorithm that penalizes the KL divergence between the marginal distributions of two domains in the representation space. Despite that this "KL guided domain adaptation" algorithm is demonstrated to outperform many existing marginal alignment algorithms [10, 11, 6, 12], it is not clear whether KL-based alignment of marginal distributions is adequate for UDA, and more fundamentally what role the unlabelled target-domain training sample should play to achieve cross-domain generalization. Notably, most UDA algorithms are heuristically designed and intuitively justified and most existing generalization bounds are algorithm-independent. Then there appears significant room for both deeper theoretical understanding and more principled algorithm design.

In this paper, we analyze the generalization ability of hypotheses and algorithms for UDA tasks using an information-theoretic framework developed in [13, 14]. The foundation of our bounding technique is the Donsker-Varadhan representation of KL divergence (see Lemma A.1) with the application of sub-gaussianity (see Assumption 2). We present novel upper bounds for two notions of generalization errors. The first notion ("PP generalization error") measures the gap between the population risk in the target domain and that in the source domain *for a hypothesis*, and the second ("expected EP generalization error") measures the gap between the population risk in the target domain and the empirical risk in the source domain *for a learning algorithm*. The specific contributions of this work

Submitted to 36th Conference on Neural Information Processing Systems (NeurIPS 2022). Do not distribute.

are as follows. We show that the PP generalization error for all hypotheses are uniformly bounded by a quantity governed by the KL divergence between the two domain distributions, which, under bounded losses, recovers the the bound in [9]. We then show that such this KL term upper-bounds some other measures including Total-Variation distance [1], Wasserstein distance [6] and domain disagreement [7]. Thus, minimizing KL-divergence forces the minimization of other discrepancy measures as well. This, together with the ease of minimizing KL [9], explains the effectiveness of the KL-guided alignment approach. For expected EP generalization error, we develop several algorithm-dependent generalization bounds. These algorithm-dependent bounds further inspire the design of two new and yet simple strategies that can further boost the performance of the KL guided marginal alignment algorithms. Experiments are performed on standard benchmarks to verify the effectiveness of these strategies.

## 2   Related Work

**Domain Adaptation**   From a theoretical perspective, many domain adaptation generalization bounds have been developed [1, 2, 15, 3, 6, 5, 7, 8], and some discrepancy measures are designed to derive these bounds including the reduction of the total variation [1, 2, 15, 3], Wasserstein distance [6], domain disagreement [7] and so on. In particular, bounds based on $\mathcal{H}\Delta\mathcal{H}$ in [2] are restricted to a binary classification setting and assume a deterministic labeling function. Furthermore, [2] also assumes the loss is the $L_1$ distance between the predicted label and true label (which is bounded). Our bounds work for the general supervised learning problems with any labelling mechanism (e.g., stochastic labelling), and we do not require the specific choice of the loss (which could be unbounded). [16] proposed some generalization bounds based on Jensen-Shannon (JS) divergence, which are related to our Corollary 4.2. Most existing works including [2, 16] that give upper bounds for Err, while we give upper bounds for its absolute value, |Err|, which also serves as a lower bound for generalization, highlighting some fundamental difficulty of the UDA learning task (see Corollary 4.1). For more details about the domain adaptation theory, we refer readers to [17] for a completed survey. From the algorithmic perspective of the domain adaptation, the most common method is to align the marginal distribution of representation between the source domain and the target domain, including using the adversarial training mechanism [10, 6, 8] and aligning the first two moments of the representation distribution [11]. There are numerous other domain adaptation algorithms, and we refer readers to [18–21] for recent advances.

**Information-Theoretic Generalization Bounds**   Information-theoretic analysis are usually used to analyze the expected generalization error of supervised learning, where the training and testing data come from the same distribution [13, 22, 14, 23–27]. By exploiting the chain rule property of mutual information, these bounds are successfully applied to characterize the generalization ability of stochastic gradient based optimization algorithms [28, 24, 26, 29–31]. Recently, this framework has also been used in the multi task setting including meta-learning [32–35], semi-supervised learning [36, 37] and some other transfer learning problems [38, 32, 39–41]. In particular, [38, 39] consider a different transfer learning problem setup with ours. Specifically, their expected generalization error is the gap between the target population risk and the empirical weighted risk (or the convex combination of the source empirical risk and the target empirical risk), while our "EP" error is the gap between the target population risk and the source empirical risk. That is to say, our work studies how to make use of the unlabelled target data to improve the generalization performance on target domain except for minimizing the empirical risk of source domain, and their works assume the training objective function for the target domain data, which could be labelled, has already been known. In addition, bounds in [38, 39] fail to characterize the dependence between $W$ and $S'_{X'}$. More precisely, the algorithm-dependent term in their bounds is $I(W; Z_i)$ or $I(W; S)$, while our algorithm-dependent term is $I^{X'_j}(W; Z_i)$ that directly depends on the unlabelled target data (see Theorem C.1 for more discussion in Appendix).

## 3   Preliminary

Unless otherwise noted, a random variable will be denoted by a capitalized letter, and its realization denoted by the corresponding lower-case letter. Consider a prediction task with instance space $\mathcal{Z} = \mathcal{X} \times \mathcal{Y}$, where $\mathcal{X}$ and $\mathcal{Y}$ are the input space and the label (or output) space respectively. Let $\mathcal{F}$ be the hypothesis space of interesting, in which each $f \in \mathcal{F}$ is a function or predictor mapping $\mathcal{X}$ to $\mathcal{Y}$. We assume that each hypothesis $f \in \mathcal{F}$ is parameterized by some weight parameter $w$ in some space $\mathcal{W}$ and may write $f$ as $f_w$ as needed.

Let $\mu$ and $\mu'$ be two distributions on $\mathcal{Z}$, unknown to the learner. Normally, $\mu$ and $\mu'$ are not the same and we consider $\mu$ characterizing the source domain and $\mu'$ characterizing the target domain. For the ease of notation, we may also write $\mu$ as $P_Z$ or $P_{XY}$ and $\mu'$ as $P_{Z'}$ or $P_{X'Y'}$, which also defines random variables $Z = (X, Y)$ and $Z' = (X', Y')$. Let $S = \{Z_i\}_{i=1}^n \sim \mu^{\otimes n}$ be a labeled source-domain training sample and $S'_{X'} = \{X'_j\}_{j=1}^m \sim P_{X'}^{\otimes m}$ be an unlabelled target-domain training sample. The objective of UDA is to design an algorithm $\mathcal{A}$ takes $S$ and $S'_{X'}$ as the input and outputs a weight $W \in \mathcal{W}$, giving rise to a predictor $f_W \in \mathcal{F}$ that "works well" on the target domain. Note that the algorithm $\mathcal{A}$ is in general characterized by a conditional distribution $P_{W|S,S'_{X'}}$.

To be precise on the performance metric of UDA, let $\ell : \mathcal{Y} \times \mathcal{Y} \to \mathbb{R}_0^+$ be a loss function. Then for each weight configuration $w \in \mathcal{W}$, its population risk in the target domain is defined as

$$R_{\mu'}(w) \triangleq \mathbb{E}_{Z'}[\ell(f_w(X'), Y')].$$

and a good UDA algorithm hopes to return a weight $w$ that minimizes this risk. Since $\mu'$ is unknown, this risk can not be measured or minimized. On the other hand, one does have access to the empirical risk in the source domain, as is defined by

$$R_S(w) \triangleq \frac{1}{n} \sum_{i=1}^n \ell(f_w(X_i), Y_i).$$

Then the notion generalization error in this setting measures how well the hypothesis returned from the algorithm generalize from the source-domain training sample to the target-domain unknown distribution $\mu'$. Taking into account the stochastic nature of the algorithm $\mathcal{A}$, a natural notion of generalization error for UDA can be defined by

$$\text{Err} \triangleq \mathbb{E}_{W,S}[R_{\mu'}(W) - R_S(W)] = \mathbb{E}_{W,S,S'_{X'}}[R_{\mu'}(W) - R_S(W)], \tag{1}$$

where the expectation in the first equation is taken over the joint distribution of $(W, S) \sim P_{W|S} \times \mu^{\otimes n}$, and the expectation of the second equation is taken over the joint distribution of $(W, S, S'_{X'}) \sim P_{W|S,S'_{X'}} \times \mu^{\otimes n} \times P_{X'}^{\otimes m}$.

Note that there is another notion of generalization error, more traditional in the domain adaptation literature, namely, the gap between the population risk in the target domain and that in the source domain, as us define by

$$\widetilde{\text{Err}}(w) \triangleq R_{\mu'}(w) - R_\mu(w). \tag{2}$$

where $R_\mu(w) \triangleq \mathbb{E}_Z[\ell(f_w(X), Y)]$. It is apparent that $\widetilde{\text{Err}}(w)$ and Err are related by the following triangle inequality:

$$|R_{\mu'}(w) - R_S(w)| \le |R_{\mu'}(w) - R_\mu(w)| + |R_\mu(w) - R_S(w)|.$$

where the second term on the right hand side is the standard generalization error in the source domain, which can be bounded by classical learning-theoretic tools, e.g., Rademacher complexity [42]. Thus bounding $\widetilde{\text{Err}}(w)$ helps bounding Err.

This paper studies both notions of generalization error for UDA. Specifically, starting from Section 5, we will mainly use information-theoretic tools to bound Err directly, without going through $\widetilde{\text{Err}}(w)$. For the ease of reference, we refer to $\widetilde{\text{Err}}(w)$ as the *population-to-population (PP) generalization error for $w$* and Err as the *expected empirical-to-population (EP) generalization error for the algorithm $\mathcal{A}$*.

Some definitions are prerequisite in this paper, we now present some uncommon notions and defer the common notions to Appendix.

**Definition 1** (Disintegrated Mutual Information). *Let $X$, $Y$ and $Z$ be random variables and $z$ be a realization of $Z$. The disintegrated mutual information of $X$ and $Y$ given $Z = z$ is $I^z(X; Y) \triangleq$* $\text{D}_{\text{KL}}(P_{X,Y|Z=z} || P_{X|Z=z} P_{Y|Z=z})$.

Note that the conditional mutual information $I(X; Y|Z) = \mathbb{E}_Z I^Z(X; Y)$.

**Definition 2** (Lautum Information [43]). *Define the lautum information between $X$ and $Y$ as* $L(X; Y) \triangleq \text{D}_{\text{KL}}(P_X P_Y || P_{XY})$.

## 4  Upper Bounds for PP Generalization Error

In this section, we present some upper bounds for $\widetilde{\mathrm{Err}}(w)$. The key techniques used in developing these bounds are the information-theoretic tools in the style of Lemma A.1. All these bounds adopt certain KL divergence as a key quantity measuring the discrepancy between the source and target domain. Notably, some previously established bounds are recovered under a different assumption of the loss function. Additionally we demonstrate that under certain conditions, the KL-based bound is an upper bound of many other discrepancy measures and hence minimizing the KL divergence forces the minimization of these other measures.

We first list some common assumptions on the loss function, which we consider in this paper.

**Assumption 1** (Boundedness). $\ell(\cdot, \cdot)$ is bounded in $[0, M]$.

**Assumption 2** (Subgaussianity). $\ell(f_w(X), Y)$ is R-subgaussian[1] under $\mu$ for any $w \in \mathcal{W}$.

**Remark 4.1.** *Note that Assumption 1 implies Assumption 2, i.e., if $\ell(f_w(X), Y)$ is bounded in $[0, M]$, then it is also $M/2$-subgaussian. Thus, Assumption 2 is weaker than Assumption 1.*

**Assumption 3** (Lipschitzness). $\ell(f_w(X), Y)$ is $\beta$-Lipschitz continues in $\mathcal{Z}$ for any $w \in \mathcal{W}$, i.e., $|\ell(f_w(x_1), y_1) - \ell(f_w(x_2), y_2)| \leq \beta d(z_1, z_2)$.

**Remark 4.2.** *Note that Assumption 1 implies Assumption 3 when $d$ is a discrete metric, i.e., if $\ell(f_w(X), Y)$ is bounded in $[0, M]$, then it is also $M$-Lipschitz under the discrete metric.*

**Assumption 4** (Triangle). $\ell(\cdot, \cdot)$ satisfies the following the triangle inequality: $\ell(y_1, y_2) \leq \ell(y_1, y_3) + \ell(y_3, y_2)$ for any $y_1, y_2, y_3 \in \mathcal{Y}$.

### 4.1  Generalization Bounds via the Subgaussian Condition

The following generalization bound is established by combining Lemma A.1 and Assumption 2, a technique developed in [14] for information-theoretic analysis of generalization.

**Theorem 4.1.** *If Assumption 2 holds, then for any $w \in \mathcal{W}$, $\left|\widetilde{\mathrm{Err}}(w)\right| \leq \sqrt{2R^2 \mathrm{D}_{\mathrm{KL}}(\mu'||\mu)}$.*

We note that this result on one hand can be turned into a generalization upper bound providing guidance to algorithm design, and on the other hand provides a lower bound of the generalization error, which highlights some fundamental difficulty of the learning task. To illustrate this, we present an corollary of Theorem 4.1, while noting that similar development can also be applied to other bounds presented later in this paper.

To that end, suppose that each $f_w$ in the model family is expressed as the composition $g \circ h$, where $h$ is a function mapping $\mathcal{X}$ to a representation space $\mathcal{T}$ and $g$ is a function mapping $\mathcal{T}$ to $\mathcal{Y}$. For any given $h : \mathcal{X} \to \mathcal{T}$, denote by $\mu_h$ the distribution on $\mathcal{T} \times \mathcal{Y}$ obtained by pushing over $\mu$ via $h$, that is, $\mu_h(t, y) = \int \delta(t - h(x)) d\mu(x, y)$, where $\delta$ is the Dirac measure on $\mathcal{T}$. Similarly, let $\mu'_h$ denote the distribution on $\mathcal{T} \times \mathcal{Y}$ obtained by pushing over $\mu'$ via $h$.

**Corollary 4.1.** *Suppose that $f_w = g \circ h$ and that Assumption 2 holds. then for any $w \in \mathcal{W}$,*

$$R_\mu(w) - \sqrt{2R^2 \mathrm{D}_{\mathrm{KL}}(\mu'||\mu)} \leq R_{\mu'}(w) \leq R_\mu(w) + \sqrt{2R^2 \mathrm{D}_{\mathrm{KL}}(\mu'_\mathrm{h}||\mu_\mathrm{h})}.$$

In this result, the lower bound of $R_{\mu'}(w)$ indicates a fundamental difficulty in UDA learning in that, using the same predictor mapping $f_w$, there is no way for the population risk in the target domain to be lower than that of the source domain less a constant which depends only on the domain difference. On the other hand, the upper bound suggests that it is possible to squeeze the gap between the two population risks by choosing an appropriate representation map $h$ - evidently such a map should be attempting to align $\mu'_h$ with $\mu_h$ or to align their respective proxies.

It is also remarkable that under Assumption 1 and due to Remark 4.1, Theorem 4.1 implies

$$\left|\widetilde{\mathrm{Err}}(w)\right| \leq \frac{M}{\sqrt{2}} \sqrt{\mathrm{D}_{\mathrm{KL}}(P_{X'}||P_X) + \mathrm{D}_{\mathrm{KL}}(P_{Y'|X'}||P_{Y|X})}. \tag{3}$$

Similarly applying this result in the representation space $\mathcal{T}$, we see that Eq. (3) recovers the bound in Proposition 1 of [9]. Notice that unlike [9], Theorem 4.1 ( or Eq. (3)) does not require the loss to be the cross entropy loss.

---

[1] A random variable $X$ is $R$-subgaussian if for any $\rho$, $\log \mathbb{E} \exp(\rho(X - \mathbb{E}X)) \leq \rho^2 R^2 / 2$.

Theorem 4.1 and [9] both use the KL divergence from source domain to target domain, $\mathrm{D}_{\mathrm{KL}}(\mu'||\mu)$, and in fact, $\left|\widetilde{\mathrm{Err}}(w)\right|$ can also be upper bounded by $\mathrm{D}_{\mathrm{KL}}(\mu||\mu')$. This can be done by invoking the subgaussianality of $\ell(f_w(X'), Y')$ (rather than $\ell(f_w(X), Y)$); for bounded loss, the subgaussianality of $\ell(f_w(X'), Y')$ is also satisfied. Then we obtain the following corollary.

**Corollary 4.2.** *If Assumption 1 holds,* $\left|\widetilde{\mathrm{Err}}(w)\right| \leq \frac{M}{\sqrt{2}}\sqrt{\min\{\mathrm{D}_{\mathrm{KL}}(\mu||\mu'), \mathrm{D}_{\mathrm{KL}}(\mu'||\mu)\}} \leq \frac{M}{2}\sqrt{\mathrm{D}_{\mathrm{KL}}(\mu||\mu') + \mathrm{D}_{\mathrm{KL}}(\mu'||\mu)}.$

**Remark 4.3.** *In the second inequality of Corollary 4.2,* $\mathrm{D}_{\mathrm{KL}}(\mu||\mu') + \mathrm{D}_{\mathrm{KL}}(\mu'||\mu)$ *is usually called the symmetrized KL divergence (or Jeffrey's divergence [44]), and the regularization term used in [9] is indeed the symmetrized KL divergence between the distributions of the source and target representations. Notice that bounds in [16] are based on the JS divergence. Since there is a sharp upper bound of the JS divergence based on Jeffrey's divergence [45], minimizing Jeffrey's divergence (in the representation space) will simultaneously penalize the JS divergence.*

In UDA, since $Y'$ is completely unavailable to the algorithm $\mathcal{A}$, it is impossible to minimize the misalignment of conditional distributions, i.e. $\mathrm{D}_{\mathrm{KL}}(P_{Y'|T'}||P_{Y|T})$, without any additional information. A common method is to assign pseudo labels to target data. However, it may also cause some additional issues. For concreteness, suppose the trained model $Q$ can well approximate the real mapping between $X$ and $Y$ on source domain (i.e. $Q_{Y|T} = P_{Y|T}$), which is usually the training objective. Let $\hat{Y}'$ be the pseudo label of $T'$ generated by the trained model, i.e., $Q_{\hat{Y}'|T'} = Q_{Y|T}$. Let $Q_{T',\hat{Y}'} = P_{T'}Q_{\hat{Y}'|T'}$, then the following holds,

$$\mathrm{D}_{\mathrm{KL}}(P_{T',Y'}||P_{T,Y}) = \mathbb{E}_{P_{T',Y'}} \log \frac{P_{T',Y'}Q_{T',\hat{Y}'}}{Q_{T',\hat{Y}'}P_{T,Y}} = \mathrm{D}_{\mathrm{KL}}(P_{T'}||P_T) + \mathrm{D}_{\mathrm{KL}}(P_{Y'|T'}||Q_{\hat{Y}'|T'}). \quad (4)$$

For a specific $t'$, if $P(Y' = y'|T' = t') \neq 0$ and $P(\hat{Y} = y'|T' = t') = 0$, then the second term in RHS of Eq. (4), $\mathrm{D}_{\mathrm{KL}}(P_{Y'|T'}||Q_{\hat{Y}'|T'}) \to \infty$. In this case, even the marginal distributions are perfectly aligned, the overall value of the upper bound is large. Thus, incorrect pseudo labels may even have negative impact on the target domain performance, and we hope two supports, $\mathrm{Supp}(P_{Y'})$ and $\mathrm{Supp}(P_{\hat{Y}'})$, could largely overlap with each other for every target data.

Indeed, the misalignment of the conditional distributions appears to be the main difficulty of UDA [1, 8]. The next corollary suggests that this difficulty may be alleviated when the loss function satisfies the triangle property, namely, Assumption 4. It can be verified that this assumption is satisfied by the 0-1 loss and square error loss; this assumption has also been considered in previous works [3, 6].

**Theorem 4.2.** *If Assumption 4 holds and let* $\ell(f_{w'}(X), f_w(X))$ *be $R$-subgaussian for any $w, w' \in \mathcal{W}$. Then for any $w$,* $\widetilde{\mathrm{Err}}(w) \leq \sqrt{2R^2\mathrm{D}_{\mathrm{KL}}(P_{X'}||P_X)} + \lambda^*$, *where* $\lambda^* = \min_{w \in \mathcal{W}} R_{\mu'}(w) + R_\mu(w).$

In this theorem, $\lambda^*$ measures the possibility of whether the domain adaptation algorithm will succeed under the oracle knowledge of $\mu$ and $\mu'$. In particular, if the hypothesis space is large enough, the minimizer $w^*$ for the "joint population risk" $R_{\mu'}(w) + R_\mu(w)$ may give rise to $R_{\mu'}(w^*) = R_\mu(w^*) = 0$. then we're likely to generalize well on the target domain. Then the KL divergence $\mathrm{D}_{KL}(P_{X'}||P_X)$ between the two $\mathcal{X}$-marginals alone bounds the PP generalization error uniformly for all $w \in \mathcal{W}$.

This theorem motivates the strategy of penalizing $\mathrm{D}_{KL}(P_{T'}||P_T)$ in the representation space to achieve better a generalization error. The next theorem suggests that such an approach also penalizes other notions of domain discrepancy, for example, domain disagreement defined in [7, Definition 1.] and serving as a key quantity in the PAC-Bayes type of domain adaptation generalization bounds [7]:

$$\mathrm{dis}(P_X, P_{X'}) \triangleq |\mathbb{E}_{W,W',X'}[\ell(f_W(X'), f_{W'}(X'))] - \mathbb{E}_{W,W',X}[\ell(f_W(X), f_{W'}(X))]|. \quad (5)$$

**Theorem 4.3.** *If* $\ell(f_{w'}(X), f_w(X))$ *is $R$-subgaussian for any $w, w' \in \mathcal{H}$, then* $\mathrm{dis}(P_X, P_{X'}) \leq \sqrt{2R^2\mathrm{D}_{\mathrm{KL}}(P_{X'}||P_X)}.$

Note that unlike [7], here we do not require the loss function to be the 0-1 loss.

## 4.2 Generalization Bounds via the Lipschitz Condition

Wasserstein distance based generalization bound are often directly connected to, or even included in, the information-theoretic bounds [46, 27]. We now present such a bound for UDA under the Lipschitz continuity assumption of the loss function.

**Theorem 4.4.** *If Assumption 3 holds, then* $\left| \widetilde{\mathrm{Err}}(w) \right| \leq \beta \mathbb{W}(\mu', \mu)$.

Note that Theorem 4.4 can be related to the KL divergence based bounds in the previous section when the Wasserstein distance is defined with respect to the discrete metric $d$. In this case, if the loss function is bounded, it is also Liptschitz continuous, and hence Theorem 4.4 applies. On the other hand, Wasserstein distance is equivalent to the total variation distance [1, 2, 15, 3], while the latter is connected to the KL divergence via Pinsker's inequality [47, Theorem 6.5] and the Bretagnolle-Huber inequality [48, Lemma 2.1]. Thus we arrive at the following result.

**Corollary 4.3.** *If Assumption 1 holds holds and let $d$ be the discrete metric, then*

$$\left| \widetilde{\mathrm{Err}}(w) \right| \leq M \mathrm{TV}(\mu', \mu) \leq M \sqrt{ \min \left\{ \frac{1}{2} \mathrm{D_{KL}}(\mu'||\mu), 1 - e^{-\mathrm{D_{KL}}(\mu'||\mu)} \right\} }.$$

The bound in Corollary 4.3 can be immediately verified to be tighter than the bound in Eq. (3).

Parallel to Theorem 4.2, if the loss function satisfies the triangle property, we may establish another bound below, which recovers a similar result in [6, Theorem 1.].

**Theorem 4.5.** *If Assumption 4 holds and $\ell(f_w(X), f_{w'}(X))$ is $\beta$-Lipschitz in $\mathcal{X}$ for any $w, w' \in \mathcal{W}$, then for any $w \in \mathcal{W}$, $\widetilde{\mathrm{Err}}(w) \leq L \mathbb{W}(P_{X'}, P_X) + \lambda^*$, where $\lambda^* = \min_{w \in \mathcal{W}} R_{\mu'}(w) + R_{\mu}(w)$.*

Unlike the bound in [6], we do not require the classification tasks to be binary in Theorem 4.5, and the loss does not need to be the $L_1$ distance.

This section may convey the following message. Since the KL divergence based bounds upper-bounds those based on other measures of domain differences, (e.g. total variation distance, domain discrepancy etc), if we penalize the KL divergence, we will also penalize those other measures. This is practically advantageous since it is usually easier and more stable to minimize the KL divergence[9].

# 5   Upper Bounds for Expected EP Generalization Error and Applications

There are two limitations in the bounds on the PP generalization error developed in the previous section and in the traditional analysis of domain adaptation. First, such bounds are independent of $w$ and hence algorithm-independent. Second, although these bounds may inspire strategies to exploit the unlabelled target sample, e.g., aligning its marginal distribution with that of the source sample in the representation space, they only provide very limited knowledge on the role that the unlabelled target sample plays in the algorithm. We now derive upper bounds for the EP generalization error, which better utilize the dependence of the algorithm output on the unlabelled target data. Applications of these bounds in designing the learning algorithms are also presented.

## 5.1   Bounds

**Theorem 5.1.** *Assume $\ell(f_w(X'), Y')$ is $R$-subgaussian under $\mu'$ for any $w \in \mathcal{W}$. Then*

$$|\mathrm{Err}| \leq \frac{1}{nm} \sum_{j=1}^{m} \sum_{i=1}^{n} \mathbb{E}_{X'_j} \sqrt{2R^2 I^{X'_j}(W; Z_i)} + \sqrt{2R^2 \mathrm{D_{KL}}(\mu||\mu')}.$$

**Remark 5.1.** *Note that the unlabelled target data plays a role in the first term of the bound. Indeed, more source and target data will reduce the first term of the bound. Specifically, moving the expectation inside the square root function by Jensen's inequality and since $Z_i \perp\!\!\!\perp X'_j$, the equations $I(W; Z_i|X'_j) = I(W; Z_i|X'_j) + I(Z_i; X'_j) = I(W; Z_i) + I(X'_j; Z_i|W)$ hold by the chain rule. The term $I(W; Z_i)$ will vanish as $n \to \infty$ and the term $I(X'_j; Z_i|W)$ will also vanish as $n, m \to \infty$.*

It is also worth mentioning that, from a practical perspective, the number of samples may have different impact on the different algorithms. For example, the second term (KL divergence) in our Theorem 5.1 can not be computed in the original space and we can only estimate it in the representation space. On the one hand, it seems that having more data will make the approximation (of KL between marginal distributions) more accurate. While on the other hand, some domain adaptation algorithms involve the pseudo labelling process, and assigning incorrect pseudo labels to the target data may even have negative impact on the target domain performance (as discussed in Section 4). In this case, having more target data will not improve the performance.

**Corollary 5.1.** *Let Assumption 1 hold. Then*

$$|\text{Err}| \leq \frac{M}{\sqrt{2}nm} \sum_{j=1}^{m} \sum_{i=1}^{n} \mathbb{E}_{X'_j} \sqrt{\min\left\{I^{X'_j}(W;Z_i), L^{X'_j}(W;Z_i)\right\}} + \frac{M}{\sqrt{2}} \sqrt{\min\left\{\mathrm{D}_{\mathrm{KL}}(\mu||\mu'), \mathrm{D}_{\mathrm{KL}}(\mu'||\mu)\right\}}.$$

*where $L^{X'_j}(\cdot;\cdot)$ is the disintegrated version of Lautum information.*

**Theorem 5.2.** *Assume $\ell$ is Lipschitz for both $w \in \mathcal{W}$ and $z \in \mathcal{Z}$, i.e., $|\ell(f_w(x),y) - \ell(f_w(x'),y')| \leq \beta d_1(z,z')$ for all $z,z' \in \mathcal{Z}$ and $|\ell(f_w(x),y) - \ell(f_{w'}(x),y)| \leq \beta' d_2(w,w')$ for all $w,w' \in \mathcal{W}$, then*

$$|\text{Err}| \leq \frac{\beta'}{nm} \sum_{j=1}^{m} \sum_{i=1}^{n} \mathbb{E}_{X'_j,Z_i} \mathbb{W}(P_{W|Z_i,X'_j}, P_{W|X'_j}) + \beta \mathbb{W}(\mu,\mu').$$

This bound is tighter than the bound in Theorem 5.1, as can be indicated by the following corollary.

**Corollary 5.2.** *Let Assumption 1 hold. Then*

$$\left|\widetilde{\text{Err}}\right| \leq \frac{M}{nm} \sum_{j=1}^{m} \sum_{i=1}^{n} \mathbb{E}_{X'_j,Z_i} \left[\mathrm{TV}(P_{W|Z_i,X'_j}, P_{W|X'_j})\right] + M\mathrm{TV}(\mu,\mu')$$

$$\leq \frac{1}{nm} \sum_{j=1}^{m} \sum_{i=1}^{n} \mathbb{E}_{X'_j,Z_i} \sqrt{\frac{M^2}{2}\mathrm{D}_{\mathrm{KL}}(P_{W|Z_i,X'_j}||P_{W|X'_j})} + \sqrt{\frac{M^2}{2}\mathrm{D}_{\mathrm{KL}}(\mu||\mu')}.$$

Notice that to recover Theorem 5.1 from Corollary 5.2, we can use Jensen's inequality to move the expectation over $Z_i$ inside the convex square root function.

## 5.2 Gradient Penalty as an Universal Regularizer

The algorithm-dependent bound in Theorem 5.1 tells us that one can reduce the expected generalization error by limiting the disintegrated mutual information $I^{X'_j}(W;Z_i)$. In the stochastic gradient based optimization algorithms, this term can be controlled by penalizing the gradient. To see this, we now consider a "noisy" iterative algorithm for updating $W$, e.g., SGLD. At each time step $t$, let the labelled mini-batch from the source domain be $Z_{B_t}$, let the unlabelled mini-batch from the target domain be $X'_{B_t}$, and let $g(W_{t-1}, Z_{B_t}, X'_{B_t})$ be the gradient at time $t$. Thus, the updating rule of $W$ is $W_t = W_{t-1} - \eta_t g(W_{t-1}, Z_{B_t}, X'_{B_t}) + N_t$ where $\eta_t$ is the learning rate and $N_t \sim \mathcal{N}(0, \sigma^2 \mathrm{I}_d)$ is an isotropic Gaussian noise. The next theorem is an application of Theorem 5.1 in this setting.

**Theorem 5.3.** *Let the total iteration number be $T$ and let $G_t = g(W_{t-1}, Z_{B_t}, X'_{B_t})$, then*

$$|\text{Err}| \leq \sqrt{\frac{R^2}{n} \sum_{t=1}^{T} \frac{\eta_t^2}{\sigma_t^2} \mathbb{E}_{S'_{X'}, W_{t-1}, S}\left[||G_t||^2\right]} + \sqrt{2R^2 \mathrm{D}_{\mathrm{KL}}(\mu||\mu')}.$$

**Remark 5.2.** *Considering a noisy iterative algorithm here is to simplify analysis. In fact it is also possible to analyze the original iterative gradient optimization method without noise injected. For example, one can follow the same development in [30, 31] to analyze vanilla SGD. In that case, there will be some additional terms in the bound, which are related to flatness of the found minima.*

Theorem 5.3 hints that to reduce the generalization error, one can restrict the gradient norm at each step. This strategy will also restrict the distance between the final output $W_T$ and the initialization $W_0$, effectively shrinking the hypothesis space accessible by the algorithm.

Indeed, adding gradient penalty can be applied to any existing UDA algorithm and it is simple but effective in practice. Later on we will show that even when the algorithm $\mathcal{A}$ does not access to any target data, in which case $I(W;Z_i|X'_j)$ reduces to $I(W;Z_i)$ and $g(W_{t-1}, Z_{B_t}, X'_{B_t})$ becomes $g(W_{t-1}, Z_{B_t})$, minimizing the empirical loss of source domain sample while penalizing gradient norm will still improve the performance. Notice that gradient penalty is also used in Wasserstein distance based adversarial training [49, 6], and their motivation is to stabilize the training to avoid gradient vanishing problem while here we use it to improve the generalization performance directly.

Notably the bound in Theorem 5.3 only depends on the size $n$ of labelled source sample and does not explicitly depend on $m$, the size of unlabelled target sample. With a more careful design, if we

consider the mutual information as the expected KL divergence of a posterior and a prior, based on $I^{X'_j}(W; Z_i)$ in Theorem 5.1, it is possible to create a target data dependent prior and derive a tighter bound based on some quantity similar to "gradient incoherence" in [24]. As this will introduce additional complexity in practice, we leave this as a future study.

## 5.3 Controlling Label Information for KL Guided Marginal Alignment

Consider instances in the representation space, $Z = (T, Y)$ and $Z' = (T', Y)$. Theorem 5.1 also encourage us to align the distributions of two domains in the representation space, as argued earlier. Then the KL guided marginal alignment algorithm proposed in [9] can be invoked here. One may notice that Theorem 5.1 uses $\mathrm{D_{KL}}(\mu||\mu')$ while [9] uses $\mathrm{D_{KL}}(\mu'||\mu)$. As already discussed in Section 4, this inconsistency can be ignored when loss is bounded (see Corollary 5.1).

Most domain adaptation algorithms aim to align the marginal distributions of two domains in the representation space. However, without accessing to $Y'$, it remains unknown if an UDA algorithm will work well since we cannot guarantee that discrepancy between conditional distribution $P_{Y|T}$ and $P_{Y'|T'}$ won't become too large when we align the marginals. In [9], the authors show that $\mathrm{D_{KL}}(P_{Y'|T'}||P_{Y|T})$ can be upper-bounded by $\mathrm{D_{KL}}(P_{Y'|X'}||P_{Y|X})$, if $I(X; Y) = I(T; Y)$. The authors then argue that penalizing the KL divergence of the marginals distributions is safe.

We now argue that in practice the condition $I(X; Y) = I(T; Y)$ can be difficult to satisfy if the cross-entropy loss is used to define the source-domain empirical risk.

By data processing inequality on $Y - X - T$, we know that $I(X; Y) \geq I(T; Y) = H(Y) - H(Y|T)$. Thus, to let $I(T; Y)$ reach its maximum, one must minimize $H(Y|T)$. On the other hand, let $Q_{Y|T,W}$ be the predictive distribution of labels in the source domain generated by the classifier. The expected cross-entropy loss for each $Z_i$ in the representation space is then

$$\mathbb{E}_{W,Z_i}\left[\ell(f_W(T_i), Y_i)\right] = \mathbb{E}_{Z_i}\left[\mathbb{E}_{W|Z_i}\left[-\log Q_{Y_i|T_i,W}\right]\right],$$

which also decomposes as [50, 51]

$$\mathbb{E}_{W,Z_i}\left[\ell(f_W(T_i), Y_i)\right] = H(Y_i|T_i) + \mathbb{E}_{T_i,W}\left[\mathrm{D_{KL}}(P_{Y_i|T_i,W}||Q_{Y_i|T_i,W})\right] - I(W; Y_i|T_i). \quad (6)$$

Then minimizing the expected cross-entropy loss may not adequately reduce $H(Y_i|T_i)$ but rather cause $I(W; Y_i|T_i)$ to significantly increase, particularly when the model capacity is large. This may have two negative effects. First, the condition $I(X; Y) = I(T; Y)$ is significantly violated, and $\mathrm{D_{KL}}(P_{Y'|T'}||P_{Y|T})$ is no longer upper bounded by $\mathrm{D_{KL}}(P_{Y'|X'}||P_{Y|X})$. As a consequence, aligning the two marginals alone may not be adequate. Second, large $I(W; Y_i|T_i)$ indicates $W$ just simply memorizes the label $Y_i$, resulting a form of overfitting and hurting the generalization performance.

The key take-away from the above analysis is that when aligning the marginals in UDA, controlling the source label information in the weights can be important to achieve good cross-domain generalization. A similar message can also be deduced from Theorem 5.1, when it is viewed in the repsentation space and noting $I^{T'_j}(W; Z_i) = I^{T'_j}(W; T_i,) + I^{T'_j}(W; Y_i|T_i)$.

To control label information, [51] proposed an approach called LIMIT. However this method is rather complicated and arguably hard to train in domain adaptation (see Appendix). We now derive a simple alternative strategy for this purpose.

Notice that $I^{T'_j}(W; Y_i|T_i) \leq \inf_Q \mathbb{E}_{T_i}\left[\mathrm{D_{KL}}(P(W|Y_i, T_i, T'_j = t'_j)||Q(W|T_i, T'_j = t'_j))\right]$, which is a simple extension of variational representation of mutual information [47, Corollary 3.1.]. Here $Q$ could be any distribution. By assuming $P = \mathcal{N}(W, \sigma^2 \mathrm{I}_d|Y_i, T_i, T'_j = t'_j)$ and taking $Q = \mathcal{N}(\widetilde{W}, \tilde{\sigma}^2 \mathrm{I}_d|T_i, T'_j = t'_j)$, we have

$$I^{T'_j}(W; Y_i|T_i) \leq \inf_Q \mathbb{E}_{T_i}\left[\mathrm{D_{KL}}(P(W|Y_i, T_i, T'_j = t'_j)||Q(\tilde{W}|T_i, T'_j = t'_j))\right] \propto ||W - \widetilde{W}||^2.$$

Thus, we may create an auxiliary classifier $f_{\widetilde{w}}$ that is not allowed to access to the real source label $Y$. In each iteration, we use the pseudo labels of target data (and source data) assigned by $f_w$ to train $f_{\widetilde{w}}$ and adding $||W - \widetilde{W}||^2$ as a regularizer in the training of $W$. The algorithm is given in the Appendix. Remarkably the regularizer here resembles "Projection Norm" designed in [52] for out-of-distribution generalization.

Table 1: RotatedMNIST and Digits Experiments. Results of baseline methods are reported from [9].

| Method | RotatedMNIST ($0^\circ$ as source domain) | | | | | | Digits | | | |
| | $15^\circ$ | $30^\circ$ | $45^\circ$ | $60^\circ$ | $75^\circ$ | Ave | $M \rightarrow U$ | $U \rightarrow M$ | $S \rightarrow M$ | Ave |
|---|---|---|---|---|---|---|---|---|---|---|
| ERM | 97.5±0.2 | 84.1±0.8 | 53.9±0.7 | 34.2±0.4 | 22.3±0.5 | 58.4 | 73.1±4.2 | 54.8±6.2 | 65.9±1.4 | 64.6 |
| DANN | 97.3±0.4 | 90.6±1.1 | 68.7±4.2 | 30.8±0.6 | 19.0±0.6 | 61.3 | 90.7±0.4 | 91.2±0.8 | 71.1±0.5 | 84.3 |
| MMD | 97.5±0.1 | 95.3±0.4 | 73.6±2.1 | 44.2±1.8 | 32.1±2.1 | 68.6 | 91.8±0.3 | 94.4±0.5 | 82.8±0.3 | 89.7 |
| CORAL | 97.1±0.3 | 82.3±0.3 | 56.0±2.4 | 30.8±0.2 | 27.1±1.7 | 58.7 | 88.0±1.9 | 83.3±0.1 | 69.3±0.6 | 80.2 |
| WD | 96.7±0.3 | 93.1±1.2 | 64.1±3.3 | 41.4±7.6 | 27.6±2.0 | 64.6 | 88.2±0.6 | 60.2±1.8 | 68.4±2.5 | 72.3 |
| KL | 97.8±0.1 | 97.1±0.2 | 93.4±0.8 | 75.5±2.4 | 68.1±1.8 | 86.4 | 98.2±0.2 | 97.3±0.5 | 92.5±0.9 | 96.0 |
| ERM-GP | 97.5±0.1 | 86.2±0.5 | 62.0±1.9 | 34.8±2.1 | 26.1±1.2 | 61.2 | 91.3±1.6 | 72.7±4.2 | 68.4±0.2 | 77.5 |
| KL-GP | 98.2±0.2 | 96.9±0.1 | 95.0±0.6 | **88.0±8.1** | **78.1±2.5** | **91.2** | 98.8±0.1 | **97.8±0.1** | **93.8±1.1** | **96.8** |
| KL-CL | **98.4±0.2** | **97.3±0.2** | **95.6±0.1** | 83.0±8.2 | 73.6±4.0 | 89.6 | **98.9±0.1** | 97.7±0.1 | 93.0±0.3 | 96.5 |

## 6 Experimental Results

We now perform experiments to verify the proposed techniques inspired by our theory in the previous section. The experimental setup follows that in [9].

**Datasets**    We select two popular small datasets, RotatedMNIST and Digits, to compare the different methods. In particular, RotatedMNIST is built based on the MNIST dataset [53] and consists of six domains with each domain containing $11, 666$ images. These six domains are rotated MNIST images with rotation angle $0^\circ, 15^\circ, 30^\circ, 45^\circ, 60^\circ$ and $75^\circ$, respectively. We will take the original MNIST dataset ($0^\circ$) as the source domain and take other five domains as target domains. Hence there are five domain adaptation tasks on RotatedMNIST. Digits consists of three sub-datasets, namely MNIST, USPS [54] and SVHN [55], and the corresponding domain adaptation tasks are MNIST→USPS (**M**→**U**), USPS→MNIST (**U**→**M**), SVHN→MNIST (**S**→**M**).

**Compared Methods**    Baseline methods are some popular marginal alignment UDA methods including **DANN** [10], **MMD** [12], **CORAL** [11], **WD** [6] and **KL** [9]. We also choose **ERM** for another baseline in which the algorithm can only access to the source domain sample during training. To verify the strategies inspired by our theory, we first add the gradient penalty to the ERM algorithm (**ERM-GP**), and we then combine gradient penalty (GP) and controlling label information (CL) with the recent proposed KL guided marginal alignment method, which are denoted by **KL-GP** and **KL-CL**, respectively.

**Implementation Details**    Most part of the implementation is based on the famous *DomainBed* suite [56]. Other settings are exactly the same with [9] and the results of baseline methods are reported directly from [9]. Specifically, each algorithm is run three times and we show the average performance with the error bar. Every dataset has a validation set, and the model selection scheme is based on the best performance achieved on the validation set of target domain during training (oracle). The hype-parameter searching process is also built upon the implementation in the *DomainBed* suite. Other details and additional experiments can be found in Appendix.

**Results**    From Table 1, we first notice that gradient penalty is able to help **ERM** to be more comparable with other marginal alignment methods. For example, on RotatedMNIST, **ERM-GP** outperforms **CORAL** and performs nearly the same with **DANN**. On Digits, **ERM-GP** outperforms **WD**. When GP and CL combined with KL guided algorithm, we can see that the performance can be further boosted. This justifies the discussion in Section 5.2 and Section 5.3.

## 7 Conclusion

Despite that the numerous learning techniques have been developed for domain adaptation, significant room exists for more in-depth theoretical understanding and more principled design of learning algorithms. This paper presents the information-theoretic analysis for unsupervised domain adaptation, where we query two notions of the generalization errors in this context and present novel learning bounds. Some of these bounds recover the previous KL-based bounds under different conditions and confirm the insights in the learning algorithms that align the source and target distributions in the representation space. Our other bounds are algorithm-dependent, better exploiting the unlabelled target data, which have inspired novel and yet simple schemes for the design of learning algorithms. We demonstrate the effectiveness of these schemes on standard benchmark datasets.

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
