# OpenReview forum: "Information-Theoretic Analysis of Unsupervised Domain Adaptation"
_NeurIPS.cc/2022/Conference — NeurIPS 2022 Submitted_

### Official Review · Reviewer_PFbK · 2022-07-06

**Rating:** 6
**Confidence:** 3
**Soundness:** 3 good
**Presentation:** 3 good
**Contribution:** 3 good

**Summary:**

This paper focuses on the generalization bound of Unsupervised Domain Adaptation (UDA). The authors discuss two kinds of generalization errors and present the upper bounds of them respectively. The theoretical analysis are based on information theory and give inspiring insights into algorithm designs. The experiments validate the effectiveness of the algorithm on benchmark datasets.

**Questions:**

Q1. Compared Theorem 5.2 with Theorem 4.4, the difference between them is the first term of the bound in Theorem 5.2. And we know that $Err$ can be bounded by $\widetilde{Err}$ plus a standard generalization error term on source domain. So, what is the difference between the first term of the bound in Theorem 5.2 and the standard generalization error term on source domain? Which one is tighter?

Other questions can be found in Cons.

**Limitations:**

Yes.

**Strengths And Weaknesses:**

## Pros:
1. Quality: This paper has adequate contribution to the generalization bound of UDA problem. They focuse on two kinds of generalization bounds. For the first error bound, their works are similar to the traditional analysis, but they generalize them that they do not require specific loss function and specific classification settings. For the second error bound, their bounds involve $W$, which motivates the algorithm designs. They derive the bounds under different assumptions and divergence measures.
2. Originality: This is the first work to analyse the generalization bounds of UDA from an information-theoretic perspective, as the authors claim. In fact, I am not familiar with the information theoretical framwork (Donsker-Varadhan representation of KL divergence) used in this paper. **So I am not very sure that the derivations in this paper are non-trival and novel.**
3. Clarity: The writing is clear and easy to follow. The paper is well-organized.
4. Significance: The generalization bounds of UDA are worth exploring. This paper is of great significance to this research field.

## Cons:
1. Some extra explanations and remarks should be added. Some contexts and symbols about information theory are somewhat hard to read.  For example, above Line 241, $I^{X_j'}(W;Z_i)$ is the disintegrated mutual information of $W$ and $Z_i$. According to the Definition 1, it involves the terms $P_{W,Z_i|X=X_j'}$ and $P_{W|X=X_j'}$. What does it mean for the probability of $W$ conditioned on an unlabeled sample $X$? It benifits for the readability of the paper if the authors add some explainations or remarks here.
2. Above Line 204, what is the distribution of $W,W'$ in $\mathbb{E}_{W,W',X}$ ?
3. Although I understand the meaning of $\ell(w,z)$, it seems has a conflic with the previous notion $\ell(f_w(X),Y)$.
4. In Theorem 5.1, 5.2, can we remove the term $\frac{1}{mn}\sum^{m}_{j=1}\sum^{n}_{i=1}$, since the samples $ \{X_j'\},\{Z_i\} $ are i.i.d. for all i,j?

---

> ### Author Response · Authors · 2022-08-02
> **To Reviewer PFbK:**
>
> Thank you very much for your positive comments. Below we discuss your comments and concerns in detail.
>
> >- Some extra explanations and remarks should be added. Some contexts and symbols about information theory are somewhat hard to read. For example, above Line 241, $I^{X_j'}(W;Z_i)$ is the disintegrated mutual information of $W$ and $Z_i$. According to the Definition 1, it involves the terms $P_{W|Z_i,X=X_j'}$ and $P_{W|X=X_j'}$. What does it mean for the probability of  $W$ conditioned on an unlabeled sample $X$? It benifits for the readability of the paper if the authors add some explainations or remarks here.
>
> **Response.** In the UDA problem setup, an learning algorithm $\mathcal{A}$ will map the input data (source data and unlabelled target data) to an output hypothesis $W$, and the randomness of this mapping is characterize by the conditional kernel density $P\_{W|S,S'\_{X'}}$. Here is the individual version, and $P\_{W|X\_j'=x\_j'}$ is indeed the distribution of the algorithm output, $W$, given a fixed unlabelled target data $x\_j'$. Recall that $I(W;Z\_i|X\_j')=\mathbb{E}\_{X\_j'}I^{X\_j'}(W;Z\_i)$, the benefit of this disintegrated technique is that it enables us to move the expectation $\mathbb{E}\_{X\_j'}$ out of the square root function in Theorem 5.1, which will make the bound tighter than using the conditional mutual information directly (i.e. $\mathbb{E}\_{X\_j'}\sqrt{I^{X\_j'}(W;Z\_i)}\leq \sqrt{I(W;Z\_i|X\_j')}$ ).
>
>
> >- Above Line 204, what is the distribution of $W,W'$ in $\mathbb{E}_{W,W',X}$?
>
> **Response.** Here $W'$ can be regarded as an independent copy of $W$, and $(W,W')\sim P^{\otimes 2}_W$.
>
>
> >- Although I understand the meaning of $\ell(w,z)$, it seems has a conflic with the previous notion $\ell(f_w(X),Y)$.
>
> **Response.** Thanks for pointing out this inconsistent notation in Theorem 5.2, we now have fixed it.
>
>
> >- In Theorem 5.1, 5.2, can we remove the term $\frac{1}{mn}\sum^{m}_{j=1}\sum^{n}_{i=1}$ since the samples ${X_j'},{Z_i}$ are i.i.d. for all i,j?
>
> **Response.** No, they can not be simply removed. Only when the algorithm satisfies some symmetric property (i.e. the output does not depend on the ordering of the points in the input), for example, the distribution $P_{W|Z_i}$ is invariant w.r.t. $i$, in which case $I(W;Z_1)=I(W;Z_2)=\cdots=I(W;Z_n)$ and they are all equal to $I(W;Z)$. To completely remove $\frac{1}{mn}\sum^{m}\_{j=1}\sum^{n}\_{i=1}$, we need to require that $P\_{W|Z\_i,X\_j'}$ is invariant w.r.t. $i,j$ and $P\_{W|X\_j'}$ is invariant w.r.t. $j$. This may not be true for all the algorithms.
>
>
> >- Compared Theorem 5.2 with Theorem 4.4, the difference between them is the first term of the bound in Theorem 5.2. And we know that $Err$ can be bounded by $\widetilde{Err}$ plus a standard generalization error term on source domain. So, what is the difference between the first term of the bound in Theorem 5.2 and the standard generalization error term on source domain? Which one is tighter??
>
> **Response.** This is a good question, and it indeed points out some direction for the future study. The first term of the bound in Theorem 5.2 can be regarded as an upper bound of the standard generalization error on source domain.
> Most of the existing works focus on the generalization performance on the target domain as this is our primary interest, there is few study on how unlabelled target data will affect the generalization performance on the source domain. The term $I^{X\_j'}(W;Z\_i)$ in Our Theorem 5.2 may provide some insights on it. In particular, if knowing a specific $x_j'$ allows us to guess the correct value of $Z\_i$ easily according to the algorithm output $W$, then the generalization performance is reduced on source domain by having the information of $X\_j'$.  In contrast, if knowing a specific $x\_j'$ makes the guessing game more difficult, then the generalization performance is improved.
>
> In addition, if we directly investigate the standard generalization error on source domain without explicitly considering the dependence between $W$ and $S'_{X'}$, we can obtain an upper bound based on $I(W;Z_i)$ (see Section D.1 in Appendix), while the first term of the bound in Theorem 5.2 is based on $I^{X_j'}(W;Z_i)$. Although $I(W;Z_i)\leq I(W;Z_i|X_j')$, without any other information, it is unclear whether $\mathbb{E}\_{X\_j'}\sqrt{I^{X\_j'}(W;Z\_i)}$ is tighter than $\sqrt{I(W;Z\_i)}$ or not. Comparing these two terms in the different algorithm settings should be an interesting direction.

---

### Official Review · Reviewer_VJaj · 2022-07-09

**Rating:** 4
**Confidence:** 4
**Soundness:** 2 fair
**Presentation:** 3 good
**Contribution:** 2 fair

**Summary:**

The paper provides two types of information-theoretic bounds for unsupervised domain adaptation. The first is the PP generalization error, which bounds the gap between the source population risk and the target population risk. The second is the expected EP generalization error that bounds the expected gap between the target population risk and the source empirical risk. The PP generalization error is connected to the KL divergence-based marginal distribution matching method. And the EP generalization error bound is algorithm-dependent and intrigues two simple approaches that improve the KL method on rotated MNIST and digits dataset.

**Questions:**

### Major Concerns

1. Equation (1) in the main paper seems problematic. $E_{W,S}[R_{\mu^\prime}(W) - R_S(W)]$ equals to $E_{W,S,S^\prime_{X^\prime}}[R_{\mu^\prime}(W) - R_S(W)]$ when $W,S$ and $S^\prime_{X^\prime}$ are independent. However, $W=\mathcal{A}(S, S^\prime_{X^\prime})$ for UDA.

2. The first equation in Appendix Line 678 is unclear to me.


### Minor concerns
1. The subguassian assumption of loss function should be related to the specific distribution, for $(X,Y)\sim\mu$ and $(X^\prime, Y^\prime)\sim\mu^\prime$. Assumption 2 is w.r.t. $\mu$, while Theorem 5.1 is w.r.t $\mu^\prime$.
2. Line 112, typo PP -> EP

**Limitations:**

Yes, the authors have claimed that this work does not touch upon the fundamental difficulty in UDA or the lower bounds of generalization errors.

**Strengths And Weaknesses:**

### Strengths
The whole paper is well-written and easy to follow. And the proposed algorithms seem to boost the performance of KL-divergence-based marginal distribution matching methods on the two datasets.

### Weakness

1. The originality and significance of the work are not obvious

The PP generalization bound related to KL divergence is actually a similar version to the JS divergence bound of [1].
Both the PP and EP bound are related to the KL divergence of the two domain distributions $\mu, \mu^\prime$. However, this term is fixed and irrelevant to the algorithm given the domain distributions. Even though the paper mentioned some insights for learning the representation via minimizing the KL divergence, no rigorous theoretical analysis is provided.
The fundamental difficulty of UDA is not studied in this paper, e.g., the theoretical validity of the pseudo label.

2. Not enough related work discussion

The paper is highly related to [2] and [3], which also studied UDA with information-theoretic tools, but no discussion is provided.

3. Experimental evaluation is not adequate

The proposed algorithms (gradient penalty/controlling label information) in fact add a regularization for learning the source domain to avoid overfitting and improve cross-domain generalization.  Similar approaches exist in previous works like [1], but no comparison is conducted.

4. Technique Concerns

See the questions in the following section.

[1] Shui, C., Chen, Q., Wen, J., Zhou, F., Gagné, C. and Wang, B., 2020. Beyond H-divergence: Domain adaptation theory with Jensen-Shannon divergence.

[2] Wu, Xuetong, et al. "Information-theoretic analysis for transfer learning." 2020 IEEE International Symposium on Information Theory (ISIT). IEEE, 2020.

[3] Jose, Sharu Theresa, and Osvaldo Simeone. "Information-theoretic bounds on transfer generalization gap based on Jensen-Shannon divergence." 2021 29th European Signal Processing Conference (EUSIPCO). IEEE, 2021.

---

> ### Author Response · Authors · 2022-08-02
> **To Reviewer VJaj:**
>
> Thank you very much for your careful reading and constructive comments. Our responses follow.
>
> >- The originality and significance of the work are not obvious
> The PP generalization bound related to KL divergence is actually a similar version to the JS divergence bound of [1].
>
> **Response.**  Thanks for pointing out this missing reference [1], which is definitely related to our work. We add this work in the related works and discuss it in our revised paper.
>
> Indeed, the key component in the RHS of the second inequality in Corollary 4.1 is called the symmetrized KL divergence (or Jeffrey’s divergence), and there is a sharp inequality [4] between Jeffrey’s divergence ($J(p,q)$) and JS divergence ($JSD(p,q)$), $JSD(p,q)\leq \ln{\frac{2}{1+\exp{[-\frac{1}{2}J(p,q)}]}}$. Thus, this matches one of the key messages in Section 4, that is, minimizing KL divergence (in the representation space) will simultaneously penalize some previous proposed discrepancy measures including total variation/1-Wassersein distance (Corollary 4.3), domain disagreement (Theorem 4.3) and JS divergence [1].
>
> In addition, unlike Theorem 1 in [1], our upper bounds (in Theorem 4.1, Corollary 4.2, Theorem 4.4 and Corollary 4.3) are derived for the absolute error $|\mathrm{Err}|$ instead of $\mathrm{Err}$, in which case the lower bound of $R_{\mu'}(W)$ can be easily induced without applying any additional techniques as Theorem 2 in [1] (see Corollary 4.1 in the revised version).
>
> >- Both the PP and EP bound are related to the KL divergence of the two domain distributions $\mu$, $\mu'$. However, this term is fixed and irrelevant to the algorithm given the domain distributions. Even though the paper mentioned some insights for learning the representation via minimizing the KL divergence, no rigorous theoretical analysis is provided. The fundamental difficulty of UDA is not studied in this paper, e.g., the theoretical validity of the pseudo label.
>
> **Response.**  Regarding the theoretical justification for computing the KL divergence in the representation space and addressing the core difficulty of UDA, we have included a new corollary (Corollary 4.1 in the revised version). Specifically, suppose that each $f_w$ in the model family is expressed as the composition $g\circ h$, where $h$ is a function mapping ${\cal X}$ to a representation space ${\cal T}$  and $g$ is a function mapping ${\cal T}$ to ${\cal Y}$. For any given $h:{\cal X} \rightarrow {\cal T}$, denote by $\mu_{h}$ the distribution on ${\cal T}\times {\cal Y}$  obtained by pushing over $\mu$ via $h$, that is, $\mu_{h}(t, y)= \int \delta(t-h(x)) d\mu(x, y)$, where $\delta$ is the Dirac measure on ${\cal T}$. Similarly, let $\mu'\_{h}$ denote the distribution on ${\cal T}\times {\cal Y}$ obtained by pushing over $\mu'$ via $h$. Thus, the bounds $R\_{\mu}(w)-\sqrt{2R^2\mathrm{D\_{KL}(\mu'||\mu)}} \le R\_{\mu'}(w) \leq R\_{\mu}(w) + \sqrt{2R^2\mathrm{D\_{KL}(\mu'\_h||\mu\_h)}}$ hold.
> In particular, the lower bound of $R_{\mu'}(w)$ indicates a fundamental difficulty in UDA learning in that, using the same predictor mapping $f_w$, there is no way for the population risk in the target domain to be lower than that of the source domain less a constant which depends only on the domain difference. On the other hand, the upper bound suggests that it is possible to squeeze the gap between the two population risks by choosing an appropriate representation map $h$ - evidently such a map should be attempting to align $\mu'_h$ with $\mu_h$  or to align their respective proxies.

---

> > ### Author Response · Authors · 2022-08-02
> > **To Reviewer VJaj (Cont.):**
> >
> > Regarding the theoretical validity of the pseudo label, although our primary study interest is marginal alignment methods (without any pseudo labelling process such as DANN), we also add a discussion about the conditional alignment with pseudo labelling. Specifically, from the upper bound in the representation space (Corollary 4.1), we notice that $\mathrm{D_{KL}}(P_{T',Y'}||P_{T,Y})$ is a key quantity to control the generalization performance on target domain. Suppose the trained model $Q$ can well approximate the real mapping between $X$ and $Y$ on source domain (i.e. $Q_{Y|T}=P_{Y|T}$), which is usually the training objective. Let $\hat{Y'}$ be the pseudo label of $T'$ generated by the trained model, i.e., $Q_{\hat{Y'}|T'}=Q_{Y|T}$. Let $Q_{T',\hat{Y'}}=P_{T'}Q_{\hat{Y'}|T'}$, then the following holds,
> > $$
> > \mathrm{D\_{KL}}(P\_{T',Y'}||P\_{T,Y})=\mathbb{E}\_{P\_{T',Y'}}\log\frac{P\_{T',Y'}Q\_{T',\hat{Y'}}}{Q\_{T',\hat{Y'}}P\_{T,Y}}=\mathrm{D\_{KL}}(P\_{T'}||P\_{T})+\mathrm{D\_{KL}}(P\_{Y'|T'}||Q\_{\hat{Y'}|T'}).
> > $$
> >
> > For a specific $t'$, if $P(Y'=y'|T'=t')\neq 0$ and $P(\hat{Y'}=y'|T'=t')= 0$, then the second term in RHS, $\mathrm{D_{KL}}(P_{Y'|T'}||Q_{\hat{Y'}|T'})\rightarrow\infty$. In this case, even the marginal distributions are perfectly aligned, the overall value of the upper bound is large. Thus, incorrect pseudo labels may even have negative impact on the performance on target domain, and we hope two supports, $\mathrm{Supp}(P_{Y'})$ and $\mathrm{Supp}(P_{\hat{Y'}})$, could largely overlap with each other for each target data. In this paper, motivated by Theorem 4.2 in our paper and Proposition 2 in [5] (in which the conditional discrepancy is upper bounded by a constant then minimizing marginal discrepancy is sufficient to reduce the generalization error),  we mainly consider marginal alignment methods instead of proposing new pseudo labelling techniques, and all of the compared baseline methods in experiments are also marginal alignment methods. We believe that the advanced pseudo labelling methods in the literature are complementary to these marginal alignment methods.
> >
> > >- Not enough related work discussion
> > The paper is highly related to [2] and [3], which also studied UDA with information-theoretic tools, but no discussion is provided.
> >
> > **Response.** Thanks for pointing out the missing reference [3], we have added more discussion on it. Both [2] and [3] consider a different transfer learning problem setup with ours, their expected generalization error is the gap between the target population risk and the empirical weighted risk (or the convex combination of the source empirical risk and the target empirical risk), while our EP error is the gap between the target population risk and the source empirical risk. That is to say, our work studies how to make use of the unlabelled target data to improve the generalization performance on target domain except for minimizing the empirical risk of source domain, and their works assume that the training objective function for the target domain data (that could be labelled), has been already known.
> >
> > In our initial submission, we have already compared our Theorem 5.1 with Corollary 2 in [2] (see Theorem C.1 in Appendix) and mention that bounds in [2] (and [3]) fail to characterize the dependence between $W$ and $S'_{X'}$. Specifically, the algorithm-dependent term in their bounds is $I(W;Z_i)$ or $I(W;S)$, while our algorithm-dependent term is $I^{X_j'}(W;Z_i)$ that directly depends on the unlabelled target data.
> >
> > >- Experimental evaluation is not adequate
> > The proposed algorithms (gradient penalty/controlling label information) in fact add a regularization for learning the source domain to avoid overfitting and improve cross-domain generalization. Similar approaches exist in previous works like [1], but no comparison is conducted.
> >
> > **Response.** We are willing to compare with the regularization methods proposed in [1], but some implementation details are not clearly provided in their paper, and it seems that [1] does not share the source code publicly (maybe it is because their work is still under submission). Indeed, as mentioned before, there are many regularization techniques (and advance pseudo labelling techniques) in the literature, and most of them (including [1]) are complementary to our method. Although our original intention is not to seek the SOTA performance and only compare with the marginal alignment methods, if the reviewer thinks there are other relevant works that we should compare with, we would be happy to conduct more empirical comparison in the revision.

---

> > > ### Author Response · Authors · 2022-08-02
> > > **To Reviewer VJaj (Cont.):**
> > >
> > > >- Equation (1) in the main paper seems problematic.$\mathbb{E}\_{W,S}[R\_{\mu'}(W)-R\_S(W)]$ equals to $\mathbb{E}\_{W,S,S'\_{X'}}[R\_{\mu'}(W)-R\_S(W)]$ when $W$, $S$ and $S'\_{X'}$ are independent. However, $W=\mathcal{A}(S,S'\_{X'})$ for UDA.
> > >
> > > **Response.** Equation (1) is correct. Suppose there are two random variables $X$ and $Y$, and let $F(X)$ be $X$-measurable. Then the equation $\mathbb{E}\_{X,Y} F(X) = \mathbb{E}\_{X} F(X)$ holds without any independence condition between $X$ and $Y$. If $X$ and $Y$ are not independent, then $X\sim P\_{X|Y=y}$ in $\mathbb{E}\_{X,Y}$ and $X\sim P\_{X}$ in $\mathbb{E}\_{X}$. If $X$ is independent of $Y$, then $X$ in both sides are drawn from the marginal distribution. Here we utilize the same concept. Notice that $R_{\mu'}(W)-R_S(W)$ is $(W,S)$-measurable, let's denote $R_{\mu'}(W)-R_S(W)$ by $F(W,S)$, then
> > > $\mathbb{E}\_{W,S,S'\_{X'}}[F(W,S)]= \mathbb{E}\_{W,S}[F(W,S)]$ without requiring any independence between random variables. Notice that here $W$ in RHS and LHS are drawn from marginal distribution and conditional distribution, respectively. More precisely, let's ignore $S$ temporarily, expectation in RHS is evaluated over the marginal distribution of $W$ (i.e. $P\_W$), while expectation in LHS is first evaluated over the conditional distribution of $W$ (i.e. $P\_{W|S'\_{X'}=s'\_{x'}}$) for every given $S'\_{X'}=s'\_{x'}$ and is then averaged over every $S'_{X'}$.
> > >
> > > >- The first equation in Appendix Line 678 is unclear to me.
> > >
> > > **Response.** The concern here may be caused by the same reason above. Recall the equations in Line 678 (Line 729 in the revised paper):
> > > $$
> > > \begin{aligned}
> > >  \left|{\mathrm{Err}}\right|=&\left|{\frac{1}{n}\sum\_{i=1}^{n}[\mathbb{E}\_{{W,Z_i}}{\ell(f\_W(X\_i),Y\_i)} - \mathbb{E}\_{{W,Z'}}{\ell(f\_W(X'),Y')}}]\right|\\\\
> > >         =&\left|\frac{1}{m}\sum\_{j=1}^{m}\mathbb{E}\_{X'_j}\left[{\frac{1}{n}\sum\_{i=1}^{n}[\mathbb{E}\_{{W,Z\_i|X\_j'=x\_j'}}{\ell(f\_W(X\_i),Y\_i)} - \mathbb{E}\_{{W,Z'|X\_j'=x\_j'}}{\ell(f\_W(X'),Y')}}]\right]\right|
> > > \end{aligned}
> > > $$
> > >
> > > The first equation is by definition and the linearity of expectation ($\mathbb{E}\_{W}[\mathbb{E}\_{\{Z\_i\}\_{i=1}^n|W=w}\frac{1}{n}\sum_{i=1}^n\ell(f\_W(X\_i),Y\_i)]=\frac{1}{n}\sum\_{i=1}^n\mathbb{E}\_{W,Z\_i}\ell(f\_W(X\_i),Y\_i)$). The second equation is by the following:
> > > $$
> > > \begin{aligned}
> > >  &\left|{\frac{1}{n}\sum\_{i=1}^{n}[\mathbb{E}\_{{W,Z\_i}}{\ell(f\_W(X\_i),Y\_i)} - \mathbb{E}\_{{W,Z'}} {\ell(f\_W(X'),Y')}}]\right| \\\\
> > >  =&\left|\frac{1}{m}\sum\_{j=1}^{m}{\frac{1}{n}\sum\_{i=1}^{n}[\mathbb{E}\_{{W,Z\_i}}{\ell(f\_W(X\_i),Y\_i)} - \mathbb{E}\_{{W,Z'}}{\ell(f\_W(X'),Y')}}]\right|\\\\
> > >  =&\left|\frac{1}{m}\sum\_{j=1}^{m}{\frac{1}{n}\sum\_{i=1}^{n}[\mathbb{E}\_{{W,Z\_i,X'\_j}}{\ell(f\_W(X\_i),Y\_i)} - \mathbb{E}\_{{W,Z',X'\_j}}{\ell(f\_W(X'),Y')}}]\right|\\\\
> > >         =&\left|\frac{1}{m}\sum\_{j=1}^{m}\mathbb{E}\_{X'\_j}\left[{\frac{1}{n}\sum\_{i=1}^{n}\mathbb{E}\_{{W,Z\_i|X\_j'=x\_j'}}{\ell(f\_W(X\_i),Y\_i)} - \mathbb{E}\_{{W,Z'|X\_j'=x\_j'}}{\ell(f\_W(X'),Y')}}\right]\right|
> > > \end{aligned}
> > > $$
> > >
> > > Since $\mathrm{Err}$ is a constant, the first equation above holds ($\mathrm{Err}=\frac{1}{m}\sum_{j=1}^m\mathrm{Err}$). Then, for each $j$, similar to the previous response, we utilize $\mathbb{E}\_{W,Z\_i,X\_j'} F(W,Z\_i)=\mathbb{E}\_{W,Z\_i} F(W,Z\_i)$ and $\mathbb{E}\_{W,Z',X\_j'} F(W,Z')=\mathbb{E}\_{W,Z'} F(W,Z')$. The last equation is due to the fact that $P\_{X\_j'}$ is irrelevant to $i$.
> > >
> > > These steps are important to understand our techniques used in Theorem 5.1, please do let us know if the reviewer still has questions.
> > >
> > > >- The subguassian assumption of loss function should be related to the specific distribution, for $(X,Y)\sim\mu$  and $(X',Y')\sim\mu'$. Assumption 2 is w.r.t. $\mu$, while Theorem 5.1 is w.r.t $\mu'$.
> > >
> > > **Response.** Our notations $\ell(f_w(X),Y)$ and $\ell(f_w(X'),Y')$ may already indicate the random variable is either $(X,Y)$-measurable or $(X',Y')$-measurable. To avoid such confusion, we have explicitly specified the distribution in the Theorem/Assumption statements.
> > >
> > > >- Line 112, typo PP -> EP
> > >
> > > **Response.** Thanks for pointing out this typo, we have fixed it.
> > >
> > > [1] Shui, C., Chen, Q., Wen, J., Zhou, F., Gagné, C. and Wang, B., 2020. Beyond H-divergence: Domain adaptation theory with Jensen-Shannon divergence.
> > >
> > > [2] Wu, Xuetong, et al. "Information-theoretic analysis for transfer learning." 2020 IEEE International Symposium on Information Theory (ISIT). IEEE, 2020.
> > >
> > > [3] Jose, Sharu Theresa, and Osvaldo Simeone. "Information-theoretic bounds on transfer generalization gap based on Jensen-Shannon divergence." 2021 29th European Signal Processing Conference (EUSIPCO). IEEE, 2021.
> > >
> > > [4] Gavin E. Crooks. Inequalities between the jenson-shannon and jeffreys divergences. In Tech.504 Note 004, 2008.
> > >
> > > [5] Nguyen, A. Tuan, et al. "KL Guided Domain Adaptation." ICLR 2022.

---

> > > > ### Comment · Reviewer_VJaj · 2022-08-07
> > > > **Response**
> > > >
> > > > Thanks for the authors' detailed reply!
> > > >
> > > > The technique point is ok for me now.
> > > > My major concerns are as follows:
> > > >
> > > > The equivalence of $E_{W,S}[R_{\mu^\prime}(W) - R_S(W)]$ and $E_{W, S, S^\prime_{X^\prime}}[R_{\mu^\prime}(W) - R_S(W)]$  means the unlabelled target data does not affect the true generalization error. While Theorem C.1 and Theorem 5.1 provide two upper bounds from these two terms, which are not compatible. And the gradient norm bound in Theorem 5.3 is not related to the number of unlabelled data. So I doubt whether the proposed theory correctly characterized the real generalization property or Theorem 5.1 is only a looser version of Theorem C.1.
> > > >
> > > > In my opinion, the experimental validation of the test performance changes with the number of the unlabeled target data is more related to the minimization of $D_{KL}(\mu_h||\mu^\prime_h)$. Moreover, to analyze the sample complexity, the bound should contain the empirical estimation of $D_{KL}(\mu_h||\mu^\prime_h)$ based on data sets, as most previous theoretical works have done.
> > > >
> > > > Finally, I acknowledge that the whole paper is mathematically solid. While I agree with Reviewer UmkN that the theorems and algorithm proposed are not surprising to me. The main novelty of the paper is adapting mutual information theory to UDA.

---

> > > > > ### Author Response · Authors · 2022-08-08
> > > > > **Response to Reviewer VJaj (1/2):**
> > > > >
> > > > > Thanks for taking the time to reply our rebuttal! We hope to address all concerns below.
> > > > >
> > > > > >- The equivalence of $\mathbb{E}\_{W,S}[R\_{\mu'}(W)-R\_S(W)]$ and  $\mathbb{E}\_{W,S,S'\_{X'}}[R\_{\mu'}(W)-R\_S(W)]$
> > > > >  means the unlabelled target data does not affect the true generalization error.
> > > > >
> > > > > **Response.** First, we would like to argue that this statement is not correct. In fact, the equivalence indicates that unlabelled target data is relevant to the generalization error. This is because the distribution of $S'\_{X'}$ affect the marginal distribution of $W$. In $\mathbb{E}\_{W,S}[R\_{\mu'}(W)-R\_S(W)]$, although unlabelled target data does not explicitly appear, to compute this expectation requires the distribution of algorithm output, $P\_W$, which will **be changed if $P\_{S'\_{X'}}$ changes**. We add Figure 1 in the revision to illustrate the Markov relationship of different random variables, from which we show that $R\_{\mu'}(W)-R\_S(W)$ depends on $S'\_{X'}$ through $W$, and all we can say is that, conditioning on $W=w$, $R\_{\mu'}(w)-R\_S(w)$ is independent of $S'\_{X'}$. It might be clearer if we replace $W$ by $\mathcal{A}(S,S'\_{X'})$,
> > > > > $$
> > > > > \mathbb{E}\_{\mathcal{A}(S,S'\_{X'}),S}[R\_{\mu'}(\mathcal{A}(S,S'\_{X'}))-R\_S(\mathcal{A}(S,S'\_{X'}))]=
> > > > > \mathbb{E}\_{S'\_{X'}}\left[\mathbb{E}\_{\mathcal{A}(S,S'\_{X'}=s'\_{x'}),S}[R\_{\mu'}(\mathcal{A}(S,s'\_{x'}))-R_S(\mathcal{A}(S,s'\_{x'}))|S'\_{X'}=s'\_{x'}]\right]
> > > > > $$
> > > > > RHS just explicitly tells us that how $P\_{S\'_{X'}}$ will change the generalization error.
> > > > >
> > > > > >- While Theorem C.1 and Theorem 5.1 provide two upper bounds from these two terms, which are not compatible.
> > > > >
> > > > > **Response.** Notice that the bound in Theorem C.1 depends on $\sqrt{I(\mathcal{A}(S,S'\_{X'});Z\_i)}$, which is indeed related to the distribution of $P\_{S'\_{X'}}$ (since computing mutual information requires $P\_W$ that is changing with $P\_{S'\_{X'}}$). The different forms of two bounds are due to the different bounding techniques. For Theorem C.1, we hope to directly obtain the Donsker and Varadhan's variational representation of the (unconditional) expectation $\mathbb{E}\_{\mathcal{A}(S,S'\_{X'}),S}[R\_{\mu'}(\mathcal{A}(S,S'\_{X'}))-R\_S(\mathcal{A}(S,S'\_{X'}))]$, while for Theorem 5.1, we first obtain the variational representation of the conditional expectation $\mathbb{E}\_{\mathcal{A}(S,S'\_{X'}=s'\_{x'}),S}[R\_{\mu'}(\mathcal{A}(S,s'\_{x'}))-R\_S(\mathcal{A}(S,s'\_{x'}))|S'\_{X'}=s'\_{x'}]$, and then take expectation over $X_j'$, which allows us to move the expectation outside of the square root function (e.g., $\mathbb{E}\_{X\_j'}\sqrt{I^{X\_j'}(W;Z\_i)}$).
> > > > >
> > > > > >- And the gradient norm bound in Theorem 5.3 is not related to the number of unlabelled data. So I doubt whether the proposed theory correctly characterized the real generalization property or Theorem 5.1 is only a looser version of Theorem C.1.
> > > > >
> > > > > **Response.** The bound in Theorem 5.3 still implicitly depends on the number of unlabelled target data since $P\_{S'\_{X'\_j}}$ will change if the number of unlabelled target data changes. It's also possible to let $\frac{1}{m}\sum_{j=1}^m$ explicitly appear in the gradient norm based bound if we characterize the sampling rule of SGD more carefully (with or without replacement sampling will give the different Markov chain and introduce additional complexity to the application of data processing inequality). The current Theorem 5.3 is applicable to any sampling rule.
> > > > >
> > > > > Theorem 5.1 is **not** a looser version of Theorem C.1. Specially, the following inequalities hold,
> > > > > $$
> > > > > \begin{aligned}
> > > > > \mathbb{E}\_{X_j'}\sqrt{I^{X\_j'}(W;Z\_i)}&\leq \sqrt{I(W;Z\_i|X\_j')},\\\\
> > > > > I(W;Z\_i)&\leq I(W;Z\_i|X\_j')
> > > > > \end{aligned}
> > > > > $$
> > > > > The first one is by Jensen's inequality, and the second one is by $$I(W;Z\_i|X\_j')=I(W;Z\_i|X\_j')+I(Z\_i;X\_j')=I(W,X\_j';Z\_i)=I(W;Z\_i)+I(X\_j';Z\_i|W)$$ and $I(X\_j';Z_i|W)\geq 0$.
> > > > > Thus, without any other information regarding the algorithm, it is unclear whether $\sqrt{I(W;Z\_i)}$  is tighter than $\mathbb{E}\_{X\_j'}\sqrt{I^{X\_j'}(W;Z\_i)}$ or not. In addition, from the algorithm design perspective, the explicit characterization of the role of $S'\_{X\_j}$ in Theorem 5.1 will inspire more implications in practice (applying similar analysis in Theorem 5.3 to Theorem C.1 can only obtain a bound based on the norm of source domain gradient).
> > > > >
> > > > >
> > > > > >- In my opinion, the experimental validation of the test performance changes with the number of the unlabeled target data is more related to the minimization of $\mathrm{D_{KL}}(\mu_h||\mu_h')$.
> > > > >
> > > > > **Response.** We agree that the number of samples is related to the minimization of KL, and this does not conflict the remark after Theorem 5.1 since minimizing KL is a part of our algorithm and is already related to the algorithm-dependent quantity, $I^{X_j'}(W;Z_i)$, in the first term of the bound (the update of $W$ is related to the minimization of KL).

---

> > > > > > ### Author Response · Authors · 2022-08-08
> > > > > > **Response to Reviewer VJaj (2/2):**
> > > > > >
> > > > > > >-  Moreover, to analyze the sample complexity, the bound should contain the empirical estimation of $\mathrm{D_{KL}}(\mu_h||\mu_h')$ based on data sets, as most previous theoretical works have done.
> > > > > >
> > > > > > **Response.** Thanks for the good suggestion! We now add Theorem B.1, Corollary B.1 and Theorem B.2 in Section B of Appendix in our revised paper.  Theorem B.1 shows the convergence rate of empirical KL based on VC-dimension. Corollary B.1 is a generalization bound based on the empirical KL. Theorem B.2 characterizes the convergence rate of empirical distribution to the real distribution in the KL sense for discrete space. Indeed, the empirical KL convergence rate is a separate research topic and, to be honest, we may only be able to give a limited analysis here for the time being.
> > > > > >
> > > > > > >-  Finally, I acknowledge that the whole paper is mathematically solid. While I agree with Reviewer UmkN that the theorems and algorithm proposed are not surprising to me. The main novelty of the paper is adapting mutual information theory to UDA.
> > > > > >
> > > > > > **Response.** Thanks for the acknowledgment. We still want to mention that the KL based bounds are not surprising may be due to the fact that they naturally reflect the intuition of the generalization property in UDA, namely, the population risks between two domains should be measured by some domain distribution discrepancy. This may not necessarily be taken as the downside of our theorems.

---

> > > > > > ### Comment · Reviewer_VJaj · 2022-08-08
> > > > > > **Response**
> > > > > >
> > > > > > Thanks for the authors' reply!
> > > > > >
> > > > > > I am not convinced. Given the expectation w.r.t. $S^\prime_{X^\prime}$, from the definition, we can say $R_{\mu^\prime}(W)$ and $R_S(W)$ are controlled by $S^\prime_{X^\prime}$, but the ***gap may not***.  For example, in Theorem 5.1, $I(W; S)$ measures the independence between $W$ and $S$, where $P_{W|S}=E_{S^\prime_{X^\prime}}P(W|S,S^\prime_{X^\prime})$.
> > > > > >
> > > > > > ***"Applying similar analysis in Theorem 5.3 to Theorem C.1 can only obtain a bound based on the norm of source domain gradient)."***
> > > > > > I do not agree. The bound in Theorem C.1 also implicitly depends on $S^\prime_{X^\prime}$ via $W$. If the algorithm has used $S^\prime_{X^\prime}$, the bound is exactly the norm gradient of all domains. And your Theorem 5.3 is obtained via $I(W; Z_i|X_j^\prime)$, which is the upper bound of $I(W; Z_i)$. That's why the first term in RHS of Theorem 5.3 is not related to the number of the unlabelled target sample.

---

> > > > > > > ### Author Response · Authors · 2022-08-08
> > > > > > > **Clarification**
> > > > > > >
> > > > > > > We notice that the confusion may be caused by that "random variable (RV)-dependent" and "RV's distribution-dependent" are not clearly stated in our response. We would like to clarify our arguments below.
> > > > > > >
> > > > > > > >-  I am not convinced. Given the expectation w.r.t. $S'_{X'}$, from the definition,...
> > > > > > >
> > > > > > > **Response.** Yes, the two random variables $R\_{\mu'}(W)$ and $R_S(W)$ are controlled by $S'\_{X'}$, and a function of random variables (e.g., $R\_{\mu'}(W)-R\_S(W)$) is also controlled by $S'\_{X'}$. We suppose the "gap" here is referred to the expected gap, $\mathrm{Err}=\mathbb{E}\_{W,S}[R\_{\mu'}(W)-R\_S(W)]$. Notice that $\mathrm{Err}$ is a constant and is not controlled by any specific value of $S'\_{X'}$. However, $\mathrm{Err}$ is $S'\_{X'}$ distribution-dependent, i.e., $\mathrm{Err}$ will be changed if $P\_{S'\_{X'}}$ changes. In Theorem C.1., yes, $I(W;S)$ measures the dependence between the random variables $W$ and $S$, and is not controlled by any specific value of random variable $S'\_{X'}$. But the quantity $I(W;S)$ itself, depends on the distribution of $S'\_{X'}$. As written by the reviewer, $P\_{W|S}=\mathbb{E}\_{S'\_{X'}} P(W|S,S'\_{X'})$, the conditional density $P\_{W|S}$ is controlled by the distribution of $S'\_{X'}$,  and if $P\_{S'\_{X'}}$ changes then $\mathbb{E}\_{S'\_{X'}} P(W|S,S'\_{X'})$ changes, giving the different value of $I(W;S)$.
> > > > > > >
> > > > > > > >- "Applying similar analysis in Theorem 5.3 to Theorem C.1 can only obtain a bound based on the norm of source domain gradient)." I do not agree. The bound in Theorem C.1 also implicitly depends on $S'_{X'}$ via $W$...
> > > > > > >
> > > > > > > **Response.** We would like to clarify that here "only obtain a bound based on the norm of source domain gradient" means that we will obtain a bound based on $\mathbb{E}\_{W\_{t-1},S}\left[||g(W\_{t-1},Z\_{B\_t})||^2\right]$ instead of $\mathbb{E}\_{S'\_{X'},W\_{t-1},S}\left[||g(W\_{t-1},Z\_{B\_t},X'\_{B\_t})||^2\right]$ in Theorem 5.3. Yes, $W\_{t-1}$ depends on $S'\_{X'}$ so $\mathbb{E}\_{W\_{t-1},S}\left[||g(W\_{t-1},Z\_{B\_t})||^2\right]$ is also $S'\_{X'}$ distribution dependent. However, from a practical perspective of UDA, it may not be easy to design a regularization term based on unlabelled target data from $||g(W\_{t-1},Z\_{B\_t})||^2$.
> > > > > > >
> > > > > > > To make this more concrete, consider SGD in an UDA algorithm that using both source domain gradient $g(W\_{t-1},Z\_{B\_t})$ and target domain gradient $g(W\_{t-1},X'\_{B\_t})$ to update $W$. Here the notation of $g$ may not be rigorous enough, $g(W\_{t-1},Z\_{B\_t})$ may contain both the classification error gradient and the marginal alignment gradient, while $g(W\_{t-1},X'\_{B\_t})$ only contains the marginal alignment gradient. In every iteration of SGD, assume a constant learning rate, $W\_t=W\_{t-1}-\lambda (g(W\_{t-1},Z\_{B\_t})+g(W\_{t-1},X'\_{B\_t}))$. To compute the gradient in practice, we need to know the current parameters $W\_{t-1}=w\_{t-1}$ and mini-batch data $Z\_{B\_t}=z\_{b\_t}$, $X'\_{B\_t}=x'\_{b\_t}$, In this case, the term
> > > > > > > $||g(w\_{t-1},z\_{b\_t})||^2$ only tells us reducing the source domain gradient norm will benefit generalization and there is no way we can explore the role of unlabelled target data (since $w\_{t-1}$ is given in every step in practice). In contrast, Theorem 5.3 can completely reveal the SGD updating process since $g(w\_{t-1},z\_{b\_t},x'\_{b\_t})$ is defined as the summation of $g(w\_{t-1},z\_{b\_t})$ and $g(w\_{t-1},x'\_{b\_t})$.
> > > > > > >
> > > > > > > In addition, the number of unlabelled target data does not explicitly appear in Theorem 5.3 because of the step $\frac{1}{m}\sum\_{j=1}^{m}I(W;Z\_i|X\_j')\leq\frac{1}{m}\sum\_{j=1}^{m}I(W;Z\_i|S'\_{X'})=I(W;Z\_i|S'\_{X'}).$ Roughly speaking, this enables us to ignore the sampling rule so we don't need to characterize whether the given $X\_j'$ is in the current mini-batch $X'\_{B\_t}$ or not for every step $t$. In fact, the quantity $I(W;Z\_i|S'\_{X'})$ is related to the number of target data. Theorem 5.3 could be further strengthened by skipping the inequality above, but, in our opinion, the additional proof complexity may not inspire more useful regularization techniques.
> > > > > > >
> > > > > > > Moreover, we want to mention a less common opinion. Yes, Theorem 5.3 is based on $I(W;Z\_i|X\_j')$ which is looser than $I(W;Z\_i)$, and we do agree that having tighter bounds is very important from a theoretical interest.  However, as argued by [6], if a looser quantity controls generalization error to a sufficient extent, then this looser quantity may provide a more general explanation of the **empirical** generalization phenomena, as the adequacy of any formally tighter bound is then tautological. Thus, when attempting to explain generalization phenomena, use the loosest bound that suffices. Our empirical results indeed evidence the usefulness of Theorem 5.3 in practice.
> > > > > > >
> > > > > > > [6] Dziugaite, Gintare Karolina, et al. "In search of robust measures of generalization." NeurIPS 2020.

---

> > > > > > > > ### Comment · Reviewer_VJaj · 2022-08-08
> > > > > > > > **Response**
> > > > > > > >
> > > > > > > > Thanks for the authors' reply,
> > > > > > > >
> > > > > > > > As far as the algorithm is the same for Theorem 3.1 and C.1, you can obtain the similar bound from C.1 as in Theorem 5.3 with gradient $E_{W, S, S^\prime_{X^\prime}}[||g(W_{t-1}, Z_{b_t}, X^\prime_{B_t})||^2]$. Because your updates is $W_t=W_{t-1} -\eta_tg(W_{t-1},Z_{B_t}, X^\prime_{B_t})$. To note here your $P_{W,S} = E_{S^\prime_{X^\prime}}P(W,S|S^\prime_{X^\prime})$, not $P_{W,S}$.
> > > > > > > > This gradient update derives from a conventional distribution matching algorithm, not from your theorem. And such an algorithm is common in practice.
> > > > > > > >
> > > > > > > > Moreover, the whole paper addresses a relatively simple and already well-studied problem: the covariate shift. My assessment is based on whether the community can learn new information from the paper. Nobody wants to spend several hours reading a paper with limited insights.
> > > > > > > >
> > > > > > > > I thank the author for adding the discussion and the bounds of empirical estimation for KL divergence. However, these basic requirements were conducted in previous works and are missing from your ***original submission***. These are not the novelty or contribution of your works. And you really add too many contents, which takes the reviewers too much additional time and effort to verify the correctness. So I suggest a resubmission.

---

> > > > > > > > > ### Author Response · Authors · 2022-08-08
> > > > > > > > > **Thanks for the reply**
> > > > > > > > >
> > > > > > > > > Although we understand the reviewer may not change the mind, we need to address some misunderstandings here.
> > > > > > > > >
> > > > > > > > > * The title of our Section 5.2 is "Gradient Penalty as an Universal Regularizer", we only claim that adding penalty for both source and target domain gradient can benefit any gradient based algorithm in UDA. Since the mutual information bound is an algorithm-dependent bound, we only want to show what insights may be obtained if we consider the specific algorithm (e.g., SGD) here. We never say the SGD update is derived by our Theorem.
> > > > > > > > >
> > > > > > > > > * Our setting is not restricted to the covariate shift. We claim that aligning marginal distributions is sufficient to minimize the generalization error via Theorem 4.2. This does not mean our theorem only meets the covariate shift case.
> > > > > > > > >
> > > > > > > > > * In the revision, we do not claim the empirical KL result is a contribution of our work since the ideas are indeed inspired by some previous works. We don't think these results can bring some additional insights and we add these theorems in Appendix only for completing the story.
> > > > > > > > >
> > > > > > > > > Again, we thank sincerely for all the constructive comments given the reviewer.

---

### Official Review · Reviewer_UmkN · 2022-07-10

**Rating:** 4
**Confidence:** 3
**Soundness:** 3 good
**Presentation:** 2 fair
**Contribution:** 2 fair

**Summary:**

This paper presents an information-theoretic analysis for unsupervised domain adaptation, where the authors considered two notions of the generalization errors in this context and present novel learning bounds. Some of these bounds recover the previous KL-based bounds under different conditions, and other bounds are algorithm-dependent that exploits the unlabelled target data, which inspired the designing of learning algorithms. The authors also demonstrate the effectiveness of these schemes on standard benchmark datasets.

**Questions:**

No.

**Limitations:**

No.

**Strengths And Weaknesses:**

Some concerns:

1. To some extend, the results in section 4 were not surprised. The upper bounds for the population risks expressed by KL divergence were well established in the transfer learning literature, say the works by Ben-David. The issue is that such bounds are usually hard to compute and applied to algorithm designs. In addition, it is also very difficult to verify high tight such bounds are. Since this paper neither presented new proving techniques nor showed any tightness results on their presented upper bounds, I may question the usefulness of such bounds for addressing the core difficulty of transfer learning problems.

2. It seems that the number of samples in both source and target training data may play important roles in domain adaptation, which is referred to as the sample complexity issue in transfer learning. However, I do not see such characterizations in the bound presented in this paper. How does more target domain samples can effect the learning algorithms as well as the performances? This may need more detailed studies and discussions.

3. The authors mentioned "This paper presents the first information-theoretic analysis for unsupervised domain adaptation". This is a bit over strong statement, and I would not say this is the first paper to present such analyses, since there are many other theoretical papers in transfer learning area that had done such kind of analyses. The paper may present some new bounds, but shall not be the first in this area.

4. The most valuable part of this paper to me is the algorithm designs in section 5. However, it seems to me that the core idea of the proposed algorithm from the theoretical analyses in to add a L2 regularization term in the gradient decent updating, which has also been applied in many previous works. I wonder if the contribution of this paper is to provide another theoretical justification for the regularization term, or indeed there are novel steps in the algorithm that were not been considered before?

5. The equation between line 105 and 106, the second term on the right hand side should be R_{\mu} instead of R_{\mu'}.

---

> ### Author Response · Authors · 2022-08-02
> **To Reviewer UmkN:**
>
> Thank you for your constructive comments. Our responses follow.
>
> >- To some extend, the results in section 4 were not surprised. The upper bounds for the population risks expressed by KL divergence were well established in the transfer learning literature, say the works by Ben-David.
>
> **Response.** To the best of our knowledge,  for measuring the domain difference and establishing upper bounds for generalization, Ben-David primarily relied on ${\cal H}\Delta {\cal H}$ divergence, a notion defined for the binary classification setting with deterministic labelling functions. It would be greatly appreciated if the reviewer may kindly point out which paper of Ben-David is referred to here, and we would be happy to discuss its connection to this work. Nonetheless it is indeed the case that generalization bounds based on KL divergence and related measures of domain difference have been established in the domain adaptation literature. Specifically, [1,2,3]. Relative to these works, the novelty of this paper is summarized as follows.
> * Bounds in [1] are completely based on the (bounded) cross-entropy loss, here we do not require the loss to be cross-entropy or bounded in our Theorem 4.1.
> * Our problem setup is different from that in [2]. The empirical risk in [2] is a convex combination of two domain empirical risks, and the definition of generalization error used in [2] is neither "PP" error or "EP" error.
> * Most existing works including Ben-David's and [3] that give upper bounds for $\mathrm{Err}$, while we give upper bounds for its absolute value, $|\mathrm{Err}|$, which also serves as a lower bound for generalization, highlighting some fundamental difficulty of the UDA learning task. An result (new Corollary 4.1), including such lower bounds, is added in the revised paper.
>
> >- The issue is that such bounds are usually hard to compute and applied to algorithm designs. In addition, it is also very difficult to verify high tight such bounds are. Since this paper neither presented new proving techniques nor showed any tightness results on their presented upper bounds, I may question the usefulness of such bounds for addressing the core difficulty of transfer learning problems.
>
> **Response.** Indeed some existing domain discrepancy measures like $\mathcal{H}\Delta\mathcal{H}$ divergence can be hard to compute, in the original space or the representation space. Efforts to resolve this difficulty include the adversarial training framework, inspired by Ben-David et al, which proposes to compute the domain discrepancy in the representation space (e.g., DANN in our experiments). However, KL divergence can be efficiently approximated in the probabilistic  representation space using certain Gaussian approximation without relying on adversarial training, which involves an additional inner-loop optimization, requires careful synchronization between the inner and outer loops, and entails instability issues. (For more implementation details regarding approximating the KL divergence, please refer to Section D in Appendix). Remarkably, in Section 4, we show that minimizing KL divergence will simultaneously minimize total variation/1-Wasserstein distance and domain disagreement (Eq. (5)) by Corollary 4.3 and Theorem 4.3. Thus regularizing KL also regularizes other domain discrepancy measures, the effectiveness of which is demonstrated in our experimental results showing that it significantly outperforms some other approaches (e.g., KL vs DANN/WD). Thus, at least from the perspective of algorithm design, we argue that the KL-based bounds are useful.
>
> In terms of addressing the core difficulty of UDA, we have included a new corollary (Corollary 4.1 in the revised version), the lower bound in which suggests that the domain difference measured by the KL divergence also governs the fundamental hardness of the UDA task. Specifically, if we restrict the source domain and the target domain to share the same predictor function, there is no way to make the population risk in target domain lower than that in the source domain less a quantity characterized precisely by the KL divergence between the two domains.
>
> Regarding the tightness of the KL-based bounds, we note that we have not been able to evaluate whether these bounds are tighter or looser than the existing bounds. This is because those bounds are usually difficult to compute or estimate. On the other hand, despite the importance of having a tighter bound, we argue that such importance is tampered if the compared bounds are much looser than the true generalization error. From a practical point of view, the usefulness of a bound is arguably more akin to whether it correctly tracks the trend of the true error and whether it may be effectively estimated and controlled in a learning algorithm. From this view point and as demonstrated in our experiments, the KL based bounds is arguably more useful than other bounds that provide little guidance in algorithm design.

---

> > ### Author Response · Authors · 2022-08-02
> > **To Reviewer UmkN (Cont.):**
> >
> > To further illustrate the usefulness, we plot the symmetrized KL divergence ($\sqrt{\mathrm{D_{KL}}(P_T||P_{T'})+\mathrm{D_{KL}}(P_{T'}||P_{T})}$ ) and the testing error in our revised paper, from which we can see the dynamic of the testing error can be well characterized by the trend of the symmetrized KL divergence.
> >
> > >- It seems that the number of samples in both source and target training data may play important roles in domain adaptation, which is referred to as the sample complexity issue in transfer learning. However, I do not see such characterizations in the bound presented in this paper. How does more target domain samples can effect the learning algorithms as well as the performances? This may need more detailed studies and discussions.
> >
> > **Response.** We agree that sample complexity is an important issue and deserves further study. We now add a remark right after our Theorem 5.1 to discuss how the number of samples affect our bounds.
> >
> > From a theoretical perspective, more source and target data will reduce the first term of the bound in Theorem 5.1. Specifically, moving the expectation inside the square root function by Jensen's inequality, and since $Z_i$ is independent of $X_j'$, the equations $I(W;Z_i|X_j')=I(W;Z_i|X_j')+I(Z_i;X_j')=I(W;Z_i)+I(X_j';Z_i|W)$ hold by the chain rule of mutual information. The first term in RHS, $I(W;Z_i)$, will vanish as $n\rightarrow\infty$ and the second term $I(X_j';Z_i|W)$ will vanish as $n,m\rightarrow\infty$.
> >
> > It is worth mentioning that, from a practical perspective, the number of samples may have different impact on the different algorithms. For example, the second term (KL divergence) in our Theorem 5.1 can not be computed in the original space and we can only estimate it in the representation space. On the one hand, it seems that having more data will make the approximation (of KL between marginal distributions) more accurate. While on the other hand, some domain adaptation algorithms involve the pseudo labelling process, and assigning incorrect pseudo labels to the target data may even have negative impact on the testing performance (see Line 189-200 in the main paper). In this case, having more target data will not improve the performance.
> >
> > To illustrate that the relation between the number of samples and the performance on target domain (e.g., having more data will improve the approximation of KL between marginal distributions), we plot the the performance evolution w.r.t. the number of target data in Figure 1 (see Section D of Appendix in our revised paper).
> >
> >
> > >- The authors mentioned "This paper presents the first information-theoretic analysis for unsupervised domain adaptation". This is a bit over strong statement, and I would not say this is the first paper to present such analyses, since there are many other theoretical papers in transfer learning area that had done such kind of analyses. The paper may present some new bounds, but shall not be the first in this area.
> >
> > **Response.** We have removed the word ''first'' to avoid such confusion. We do notice that in some other transfer learning setting, there exist some works based on information-theoretic analysis such as [2] (and we do discuss them in the related works and make more comparison with [2] in Section C.1 in Appendix), while our analysis is the first one to specifically characterize the UDA setting.

---

> > > ### Author Response · Authors · 2022-08-02
> > > **To Reviewer UmkN (Cont.):**
> > >
> > > >- The most valuable part of this paper to me is the algorithm designs in section 5. However, it seems to me that the core idea of the proposed algorithm from the theoretical analyses in to add a L2 regularization term in the gradient decent updating, which has also been applied in many previous works. I wonder if the contribution of this paper is to provide another theoretical justification for the regularization term, or indeed there are novel steps in the algorithm that were not been considered before?
> > >
> > > **Response.** We do mention that such gradient penalty has been used in the previous works such as Wasserstein distance guided domain adaptation method [4]. However, to the best of our knowledge, none of the previous works has explained why this gradient penalty can improve the generalization performance on the target domain, and they usually interpret that adding this term would help to stabilize the adversarial training process. Our Theorem 5.3 provides a theoretical justification for such a gradient penalty term, and even in the ERM algorithm (which only uses source domain data), this regularization term can non-trivially boost the performance (as shown in Table 1).
> > >
> > > While adding gradient penalty does not involve any novel steps, another approach, controlling label information, is indeed novel (see algorithm 1) and the most similar method is [4] which is much more complicated. Furthermore, controlling label information is not equivalent to weight decay, and experimental results show that it outperforms weight decay as weight decay is a standard training component and has already been used in almost every baselines by default in our paper.
> > >
> > >
> > > >- The equation between line 105 and 106, the second term on the right hand side should be R_{\mu} instead of R_{\mu'}.
> > >
> > > **Response.** Thank you very much for pointing out this typo, we have fixed it.
> > >
> > > [1] Nguyen, A. Tuan, et al. "KL Guided Domain Adaptation." ICLR 2022.
> > >
> > > [2] Wu, Xuetong, et al. "Information-theoretic analysis for transfer learning."  ISIT 2020.
> > >
> > > [3] Shui, Changjian, et al. "Beyond $\mathcal{H}$-Divergence: Domain Adaptation Theory With Jensen-Shannon Divergence." arXiv preprint arXiv:2007.15567 (2020).
> > >
> > > [4] Shen, Jian, et al. "Wasserstein distance guided representation learning for domain adaptation." AAAI 2018.

---

> > > ### Comment · Reviewer_UmkN · 2022-08-10
> > > **Thanks for the responsed from the authors**
> > >
> > > First of all, the responses of the authors are acknowledged and appreciated. I am particularly curious on the sample complexity issues in transfer learning which are considered as a key element in theoretical analyses for this topic. The response from the authors only mentioned the asymptotic case, i.e., n,m grow to infinity. In practice, we need to consider the phenomenon of finite sample cases and characterize the theoretical bounds w.r.t. the finite number of samples in detailed ways, which I do not see in this paper. Please correct me I missed something here.
> > >
> > > I do have some attempts to increase the score, but still not quite sufficient to convince myself.
> > >
> > > Anyway, thanks for the authors for the detailed responses.

---

> > > > ### Author Response · Authors · 2022-08-10
> > > > **Regarding the Sample Complexity Issues**
> > > >
> > > > Thanks for the reply!
> > > >
> > > > We also provide some sample complexity bounds to characterize the convergence rate of the empirical KL divergence in the Appendix of the latest revision. Please check Theorem B.1 and Corollary B.1 in Section B.8 of Appendix and Theorem B.2 in Section B.9 of Appendix. Unlike Theorem 5.1 that is based on the general UDA algorithm, these sample complexity bounds are given to support the effectiveness of the KL divergence guided domain adaptation algorithms (i.e. minimizing KL in the representation space, which is the foundation method in our experiments), that is, KL divergence of two domains can be measured from finite samples from the distributions.
> > > >
> > > > For the first term in Theorem 5.1, as mentioned in the previous response, it is upper bounded by $\frac{1}{n}\sum_{i=1}^n\sqrt{I(W;Z_i)}+\frac{1}{nm}\sum_{i=1}^n\sum_{j=1}^m\sqrt{I(X_j;Z_i|W)}$, where the first term has the decaying rate $\frac{1}{\sqrt{n}}$ and the second term decays as $\frac{1}{\sqrt{n}}$ and $\frac{1}{\sqrt{m}}$. It is thoughtless of us to miss  the detailed finite sample analysis in the last revision, and even for the asymptotic case, we should mention that having infinite target data may make the first term in the bound of Theorem 5.1 smaller but will not let the term vanish.
> > > >
> > > > Although our intention may not focus on the sample complexity in the original submission, we always welcome any feedback of the reviewer regarding this issue.

---

### Official Review · Reviewer_ch1K · 2022-07-10

**Rating:** 4
**Confidence:** 4
**Soundness:** 2 fair
**Presentation:** 3 good
**Contribution:** 2 fair

**Summary:**

This paper provides two upper bounds for two notions of generalization errors; this first is the gap between the population risk in the target domain and the population risk in the source domain; the second is the gap between the population risk in the target domain and the empirical risk in the source domain. In addition, the paper presents two techniques for improving generalization in UDA and validate them experimentally.

**Questions:**

- The upper bounds seem very loose. What is the first upper bound's superiority to the previous bounds mentioned in [2]?

**Limitations:**

The limitations of the presented analysis could be discussed in more detail.



**Strengths And Weaknesses:**

### Originality

The paper's idea is not extremely novel, since there have been a lot of upper bounds for the first gap, such as [2].

### Quality

The technical part of the paper seems correct as far as I can see.

### Clarity

The paper is well-written.

### Significance

The question studied in the paper is significant. The generalization gap bounds inspire insights into algorithm designs.

---

> ### Author Response · Authors · 2022-08-02
> **To Reviewer ch1K:**
>
> We thank you sincerely for your comments to our paper. Our responses follow.
>
> >- The paper's idea is not extremely novel, since there have been a lot of upper bounds for the first gap, such as [2].
>
> **Response.** The novelty of this work may be summarized as follows.
>
> * In the context of UDA, existing literature on generalization bounds such as [2] primarily focuses on the "PP bounds", bounding the population risk gap between the source and target domains. This paper goes beyond this setting and also studies the "EP bounds", which bound the gap between the population risk on the target domain and the empirical risk on the source domain (using novel techniques beyond breaking the gap into a PP gap and the generalization gap in the source domain and invoking the triangle inequality).
> * For the "PP" bounds, only Theorem 4.2 and Theorem 4.5 are partly inspired by the classic analysis in [2], e.g., we both use $\lambda^*$ to characterize the possibility of whether the DA algorithm will succeed under the oracle knowledge of $\mu$ and $\mu'$, the remaining analysis is independent of the work of [2].
> * In addition, compared with many existing works that give upper bounds for $\mathrm{Err}$, we give upper bounds for its absolute version, $|\mathrm{Err}|$, which will naturally induce the lower bounds of the generalization error (the lower bound has been added in Corollary 4.1 in our revision).  More discussions on the comparison of our bounds with bounds in [2] are given below.
>
> >- The upper bounds seem very loose. What is the first upper bound's superiority to the previous bounds mentioned in [2]?
>
> **Response.** From a theoretical perspective, our bounds are more general and require weaker assumptions. Bounds in [2] are restricted to a binary classification setting and assume a deterministic labeling function. Furthermore, they also assume the loss is the $L_1$ distance between the predicted label and true label (which is bounded). Our bounds work for the general supervised learning problems with any labelling mechanism (e.g., stochastic labelling), and we  do not require the specific choice of the loss (which could be unbounded). From a practical perspective, even in the representation space, computing the $\mathcal{H}\Delta{\mathcal H}$ divergence in [2] is usually intractable, while computing the marginal KL divergence in the probabilistic representation space is a much easier task (as shown in Section D.1 in Appendix). Moreover, since we demonstrate that minimizing KL divergence necessarily minimizes total variation distance/1-Wasserstein distance (Corollary 4.3) and domain disagreement (Theorem 4.3), KL based bounds have more practicability; this is also indicated in experiments (e.g., DANN vs KL).

---

### Author Response · Authors · 2022-08-02
**To all reviewers:**

We would like to thank all reviewers for your insightful comments. We have revised the paper to address these comments and we will discuss them separately in our response to each reviewer. The major revised parts are summarized below (which are also highlighted in the paper).
* We add some missing references pointed out by reviewers, and we add more discussions regarding the comparison with previous works in both main paper and Appendix.
* Discussions on the limitation of this work and some backgrounds including the definition of Wasserstein distance/total variation distance and Donsker-Varadhan representation of KL divergence are moved to Appendix due to the limited space.
* A new Corollary 4.1 and some discussions are given to theoretically justify the effectiveness of minimizing KL divergence in the representation space and provide some intuition of the fundamental difficulty of the UDA problem (e.g. a lower bound of the generalization error).
* A remark about the relation between Jeffrey’s divergence and Jensen-Shannon divergence is given after Corollary 4.2.
* We add discussions regarding the role that pseudo labels play in some UDA algorithms after Remark 4.3.
* We add some remarks regarding how the number of samples theoretically and practically affect the performance of the UDA algorithms after Theorem 5.1.
* To further illustrate the usefulness of our bounds and the impact of the number of unlabelled target data, we add Figure 2 in Appendix.

---

> ### Author Response · Authors · 2022-08-08
> **To all reviewers (cont.)**
>
> * As suggested by Reviewer VJaj, we now add Theorem B.1, Corollary B.1 and Theorem B.2 in Section B of Appendix in our revised paper to analyze the convergence rate of empirical KL.

---

### Meta-Review · Area_Chair_edep · 2022-08-24

**Recommendation:** Reject
**Confidence:** Certain

**Metareview:**

The reviewers agreed that the paper's novelty is limited and lacks proper justification. The extensive authors' discussion shows that they could improve the manuscript, but the number of modifications in the revised version would require another round of reviews.

Therefore, I encourage the authors to submit an improved version of their work to an upcoming venue.

**Award:**

No

---

### Decision · Program_Chairs · 2022-09-14

Reject